# Query Design for Crowdsourced Clustering:
# Effect of Cognitive Overload and Contextual Bias

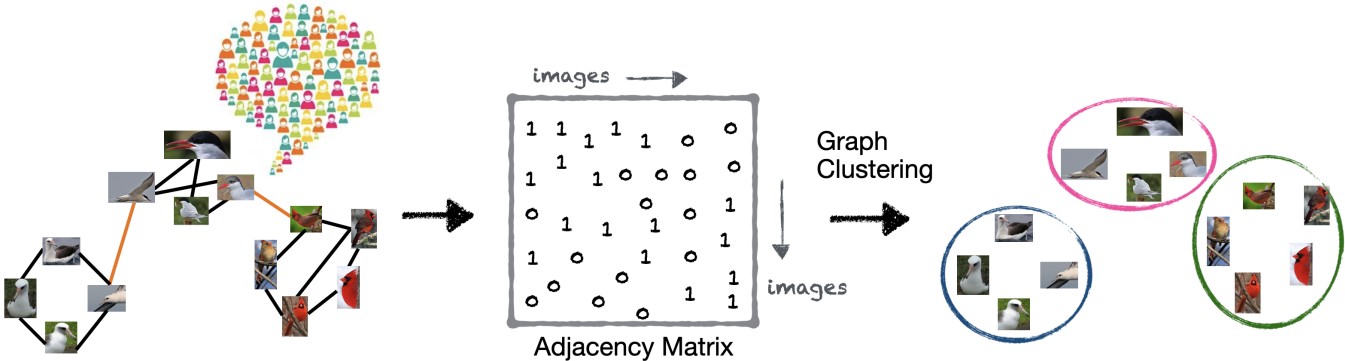

Figure 1: Pictorial description of the crowdsourced clustering system.

## Abstract

Crowdsourced clustering leverages human input to group items into clusters. The design of tasks for crowdworkers, specifically the number of items presented per query, impacts answer quality and cognitive load. This work investigates the trade-off between query size and answer accuracy, revealing diminishing returns beyond 4-5 items per query. Crucially, we identify contextual bias in crowdworker responses – the likelihood of grouping items depends not only on their similarity but also on the other items present in the query. This structured noise contradicts assumptions made in existing noise models. Our findings underscore the need for more nuanced noise models that account for the complex interplay between items and query context in crowdsourced clustering tasks.

## CCS Concepts

• **Human-centered computing** → **Empirical studies in collaborative and social computing**; • **Computing methodologies** → Cluster analysis; • **Information systems** → **Crowdsourcing**.

## Keywords

crowdsourcing, crowdsourced clustering

**ACM Reference Format:**
Anonymous Author(s). 2018. Query Design for Crowdsourced Clustering: Effect of Cognitive Overload and Contextual Bias. In *Proceedings of Make sure to enter the correct conference title from your rights confirmation emai*

*(Conference acronym 'XX).* ACM, New York, NY, USA, 23 pages. https://doi.org/XXXXXXX.XXXXXXX

## 1 Introduction

Deep neural networks, from LeNet-5 [23] to ResNet [14], have become indispensable, particularly after the unprecedented results of AlexNet [21] in the 2012 ImageNet Large Scale Visual Recognition Challenge [38]. The collection of large volumes of labeled data is a critical step that has contributed significantly to the success of these models. For example, ImageNet [10] contains 3.2 million labeled images, a massive leap from the scale of previous datasets, such as Caltech 256 [12], which had approximately 31,000 labeled images.

One way to build such a large labeled dataset is hiring experts or training ad hoc experts for the labeling task. However, this costs money and time. Instead, researchers can take advantage of the wisdom of crowds to solve this problem, that is, resort to *crowdsourcing* [37, 41], a method in which a crowd of nonexperts, known as crowdworkers, assumes the labeling task. One may argue that auto-labeling [48] or foundation model [36] can be used to label a dataset. However, in the case where a highly specialized dataset is required for a domain-specific task, utilizing crowdsourcing is inevitable.

One aspect of crowdsourcing that cannot be overlooked is that non-expert crowdworkers on platforms like Amazon Mechanical Turk (AMT) [1] provide noisy answers. However, label quality is crucial for the effectiveness of supervised algorithms. Therefore, it is vital to formulate crowdsourcing tasks that are easy for nonexperts to answer, minimizing noise, and maximizing information gathered under a given budget.

### 1.1 Crowdsourced Clustering

While it is straightforward for nonexperts to label general classes like "dog" or "cat", obtaining a more granular label, i.e. the specific breed of a given animal, is challenging.

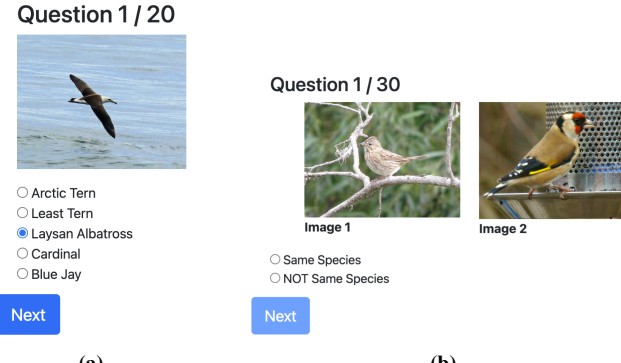

**Figure 2: Two types of queries: (a) direct labeling task and (b) comparison task.**

Consider the task of labeling images of different bird species. To accurately label each bird with its specific species (Figure 2a), a worker needs a certain level of expertise in identifying various bird species, or they need to be trained—both of which are expensive options. However, if we present a pair of bird images (Figure 2b) and ask "*Are these two birds from the same species?*", the task becomes considerably simpler than direct labeling. These pairwise comparison queries are the building blocks of crowdsourced clustering systems [11, 44, 47, 50].

A typical crowdsourced clustering system builds a graph from the answers to the queries. Given a dataset consisting of $n$ items, we construct a graph by treating each of the items in the dataset as a node. The graph is represented by an adjacency matrix $A$. Each column and row of $A$ corresponds to an item in the dataset, and the entry $i, j$ of $A$, denoted as $A_{ij}$, is an indicator of whether there is an edge between the item $i$ and the item $j$ in the graph. That is, $A_{ij} = 1$ if there is an edge between the item $i$ and the item $j$ and $A_{ij} = 0$ otherwise. At the same time, having an edge between the item $i$ and the item $j$ indicates that they are similar. Each entry, whether containing a numerical value of 0 or 1, could be in two states: observed or unobserved.

When a crowdworker decides that the item $i$ and the item $j$ are from the same cluster, the corresponding entry $A_{ij}$ is observed and filled with 1. Otherwise, the entry $A_{ij}$ is observed and filled with 0. After the querying process is done, the matrix $A$ is fed to a graph clustering algorithm, which outputs the clustering of the dataset. To obtain the labels, we hire experts to label each cluster found.

A natural question we ask is "How should we design a query?" Querying all entries is prohibitive since the number of queries is in the order of $O(n^2)$, where $n$ is the number of items in the dataset. Therefore, we need to determine what to query. Should we query two images at a time, thus filling one entry, or should we present three images to fill three entries, or four images to fill six entries of the adjacency matrix? In other words, should we present as many images as possible per query to maximize the number of edges explored and obtain better clustering? Or will the information demands of these tasks exceed the processing capabilities of the crowdworkers (*cognitive overload*)?

In addition to query design, we investigate the noise present in the answers provided by the crowdworkers. The noise level in the answers could be influenced not only by the abilities of the workers and the task difficulty but also by the specific set of items being queried, as human perception is highly context-dependent (*contextual bias*).

**Our contribution:** We conducted experiments on AMT to empirically assess the impact of *cognitive overload* and *contextual bias* in crowdsourced clustering with multi-item queries. Our findings demonstrate that the benefit of incorporating more items per query diminishes after around 4-5 items, likely due to cognitive overload. We also reveal a structured pattern in the "noise" of crowdworker responses, where item grouping depends not only on pairwise similarity but also on the broader query context and hierarchical relationships among the items. Additionally, we conduct simulations to investigate the impact of query size on clustering results, as well as address a gap in the literature focused primarily on pairwise and triangle queries.

## 2 Related Works
## 2.1 Direct Labeling Query

Many works on the theoretical understanding of crowdsourcing focus on labeling tasks, where crowdworkers are asked to label items directly [9, 13, 17–19, 24, 34, 40**?** **?** ]. [17, 18] adopted the "spammer-hammer model", which treats workers as a mixture of "spammers", who randomly answer the questions, and "hammers", who answer correctly. [24, 34] utilized methods from information theory and coding theory to reconstruct the labeling from the answers to the queries. [24] modeled noises from the crowdworker similar to bit flipping. [34] considered noise as whether a query is answered or not. [13] pointed out that although crowdworkers give incorrect errors, some are more correct than others.

## 2.2 Comparison Query

Another line of work focuses on the comparison query, where crowdworkers are asked to group the items by their similarity, which is based on crowdworkers' perception of them [2, 22, 25, 31, 42, 43, 45, 47]. [11] showed that the wisdom of crowds can be used for crowd clustering. [47] studied clustering algorithms that work with partially observed graphs and provided theoretical guarantees on when clustering works in such scenarios. [31] introduced a framework of using pairwise comparison comparison with Elo scoring to reduce the variability and bias introduced by subjectivity. [2] considers the clustering task over texts instead of images.

Methods in [25**?** ] tried to actively select images to be queried. [43] present active crowdclustering, which does not rely on any unknown parameters and can recover clusters regardless of their sizes. [6] extends this work by implementing the algorithm and conducting experiments on AMT.

[22] assumed that crowdworkers do not make mistakes, making their method less practical. The method proposed in [45], known as random triangle query, builds on top of [42] with a modification on how the question is asked. To model the noises, the authors present the conditional block model, which builds on top of the stochastic block model.

## 2.3 Cognitive Overload

The effect of cognitive overload has been studied extensively in the field of social psychology and information seeking [5, 7, 16, 33]. [33] and [5] discuss cognitive overload as a *"Less is More Effect"* in which people find it more difficult to draw comparisons when confronted with a large number of options. [16] study the effect in the setting of consumer behavior. The authors have found that consumers prefer to purchase from a vendor that displays fewer options. [7] identifies 4 key factors that impact the effect of cognitive overload via meta-analysis in the field of consumer psychology.

## 2.4 Contextual Bias

Contextual bias is the "noise" within the answers provided by crowdworkers that is a function of the set of items the crowdworkers are exposed to in a query. [2] shows that having context introduced in the task is beneficial. Yet, the authors did not investigate how much context should be added.

Both [29] and ours work try to answer the question of how the breadth of data affects the outcome of the model's result. In our work, however, the breadth concerns the set of items being shown to crowdworkers, rather than being used by the model. For the granularity aspect, [29] considers granularity as the level of detail used (by the model) to explain a model's decision. Conversely, we treat granularity as the level of detail used by crowdworkers to make their decision.

## 3 Crowdsourcing Study

### 3.1 Definitions

We refer to a *human intelligence task (HIT)* as a question that needs human answer. Each HIT consists of multiple sub-tasks, to which we refer as *query*.

### 3.2 Crowdsourced Clustering

Given a dataset of $n$ items, we crowdsource it to $n_p = 300$ unique crowdworkers that have more than 500 HITs approved and a HIT approval rate greater than 95% on AMT. Each HIT consists of $n_q$ queries. Each query presents crowdworkers $m$ ($m \ll n$) items. When a crowdworker accepts the task, our crowdsourced clustering system selects $m \cdot n_q$ items from the dataset uniformly at random. We present $n_q$ queries each with $m$ items. Queries in the task present these items sequentially to crowdworkers. That is, the first $m$ items are shown in the first query; the second $m$ items are shown in the second query, etc. Each query requires the crowdworkers to compare the $m$ items and group them by their similarity. When a query is answered, the corresponding $\binom{m}{2}$ entries in the adjacency matrix are filled. After all HITs are completed, we apply spectral clustering [32, 35, 39] with the number of clusters $K$ equal to the true number of clusters on the adjacency matrix to obtain the clustering.

Since there are $\binom{n}{m}$ possible $m$-item queries, with a limited number of queries that we can make on a budget, when we query a random subset of $m$-item queries, the probability that two crowdworkers work on exactly the same set of queries is very low. Note that we do not repeat a query multiple times, seeking to denoise the answers. This choice is informed by previous studies [44], which have shown that for clustering partially observed graphs under a

- Thank you for your interest!
- You will be shown 20 questions of images with birds in them.
- For each question, you will see 5 radio buttons.
- For the 3 images shown to you,
  - If you think birds in the 3 images are of the same species, then click on `All are Same Species`
  - If you think only birds in Image 1 and 2 are from the same species, then click on `ONLY 1 and 2 are Same Species`
  - If you think only birds in Image 1 and 3 are from the same species, then click on `ONLY 1 and 3 are Same Species`
  - If you think only birds in Image 2 and 3 are from the same species, then click on `ONLY 2 and 3 are Same Species`
  - If you think all are different species, then click on `NONE, all are different species`
- You need to answer all the questions.
- Of the 20, there are 5 (random out of 20) GOLD STANDARD questions. You need to get at least 3 of them correct to get the answers accepted.

Question 1 / 20

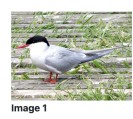 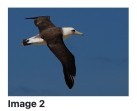 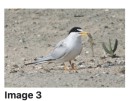

Image 1   Image 2   Image 3

○ All are Same Species
○ ONLY 1 and 2 are Same Species
○ ONLY 1 and 3 are Same Species
○ ONLY 2 and 3 are Same Species
○ NONE, all are different species

Next

**Figure 3: Samples of the radio interface deployed on Amazon Mechanical Turk with $m = 3$ and Birds5 dataset. Instructions are always shown on top of the page. Crowdworkers click on the Next button to proceed to the next query.**

given budget, the benefit of covering more edges outweighs the benefit of marginally reducing noise in the entries from repeated querying.

While filling out the entries of the adjacency matrix from multi-item queries, it is possible for an entry to be observed multiple times. In that case, we randomly pick an answer. From our empirical study, we observe that the percentage in which an edge is queried 3 times or more is less than 1%. There is no significant difference between our choice and majoirty voting in terms of selecting repeated edges.

### 3.3 Cost

We conducted a pilot experiment for both interfaces with $m = 2$ using the Dogs3 dataset. The average time for 20 crowdworkers to complete 30 drag-and-drop queries was around 201.72 ± 97.91 seconds, whereas the average time to complete 30 radio queries was approximately 122.54 ± 47.99 seconds. The reward for each drag-and-drop task was set to \$0.50, or approximately \$8.92 per hour. The reward for each radio-button task was \$0.30, or approximately \$8.81 per hour.

### 3.4 Time span

We conducted our experiments in June 2023 and in January, February, March, April, and June 2022. To save the budget, each batch of HITs we created on AMT consists of only 9 HITs. We created a series of batches for each experiment, totaling 300 HITs, and used the same uniqueturker ID to try to prevent the cases where a crowdworker works on our HITs multiple times. A batch of HITs is completed within one day of creation.

### 3.5 Interface Design

In this section, we describe how the two interfaces work. We defer the technical details of the implementation of the interfaces to Appendix A.

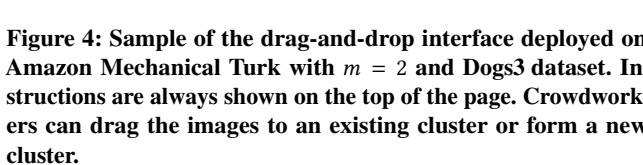

**Figure 4: Sample of the drag-and-drop interface deployed on Amazon Mechanical Turk with $m = 2$ and Dogs3 dataset. Instructions are always shown on the top of the page. Crowdworkers can drag the images to an existing cluster or form a new cluster.**

*Radio-button Interface.* We included a modified radio-button interface, similar to that used in [45], to maintain continuity with prior work. However, our version presents queries one at a time (Figure 3), allowing us to track individual query completion times. The task instructions remain at the top of the page and a progress indicator shows the number of remaining queries. Crowdworkers can provide feedback upon completion. A limitation of this interface is scalability. As $m$ increases, the number of possible groups to be shown as radio buttons increases, potentially overwhelming crowdworkers. Figure 14 in the Appendix illustrates this issue.

*Drag-and-Drop Interface.* To overcome the scaling limitations of the radio-button interface, we developed a drag-and-drop interface (Figure 4). The images are presented at the top, and the crowdworkers create groups by dragging and dropping them to the desired locations. At the end of the HIT, feedback can be submitted to us. To help crowdworkers familiarize with the interface, we introduce a tutorial and practice stage to the crowdworkers. Details regarding the tutorial and practice stage are deferred to Appendix B.

## 3.6 Datasets

Following previous work [45], we use the following datasets:
**Dogs3:** consists of images of 3 breeds of dogs from the Stanford Dogs dataset [20]: Norfolk Terrier (172), Toy Poodle (150), and Bouvier des Flandres (151), totaling 473 dog images. Figure 5 (a) - (c) displays selected images from each breed in the Dogs3 dataset.
**Birds5:** consists of images of 5 species of birds from the CUB-200-2011 dataset [49]: Laysan Albatross (60), Least Tern (60), Arctic Tern (58), Cardinal (57), and Green Jay (57). Additionally, it includes 50 random species acting as outliers, resulting in 342 bird images.
**Birds5+:** is used to investigate the effect of contextual noise. It

is constructed manually by adding 20 Common Terns from the CUB-200-211 dataset to Birds5 dataset. While selecting these Terns, we ensure the birds in these images are standing, allowing us to minimize noise associated with varied bird postures. Figure 5 (d) - (i) showcase selected bird images from each species in Birds5 and Birds5+ dataset.

## 3.7 Evaluation Metric

We use *Variation of Information* (VI) [26] to quantify clustering accuracy. VI is a metric that compares two clustering results on the same dataset. A smaller VI value indicates a closer match between the two clusterings, and a VI value of 0 denotes an identical clustering result. We also gather *worker edge error rate*: the edge error rate of each individual crowdworker.

## 3.8 Effect of Cognitive Overload in Multi-item Queries

Research conducted by Vinayak and Hassibi [45] suggests that increasing the number of items $m$ in each query would theoretically improve the performance of clustering. However, we hypothesize that when $m$ exceeds a certain threshold, we would observe a phenomenon of diminishing returns due to the cognitive overload imposed on the crowdworkers. To empirically test our hypothesis, we experiment on AMT with $m \in \{2, 3, 4, 5, 6, 7, 8\}$. Although we cannot fully ensure that crowdworkers in different experiment settings experience a similar workload, we try our best to balance the workload and budget of each experiment by fixing the total number of images shown in a HIT by 60 [1].

## 3.9 Effect of Contextual Bias

We hypothesize that errors made by crowdworkers depend not only on the design of the query but also on the items being queried, as crowdworkers may classify images based on different similarity perception hierarchy induced by the query context. To investigate this contextual bias effect, we manually select a set of items, among which some, despite belonging to different ground-truth clusters, are more similar to each other than to the rest.

We designed three experiments to illustrate the effect of contextual bias. For each of these three experiments, we ask 50 crowdworkers to complete 20 queries, where each query involves 3 images of different bird species. Across the 50 crowdworkers, we fix the (unordered) set of images presented in each query. The only difference between the experiments was the three species shown per query:
**Experiment 1** (*lt-at-ct*): Least Tern (*lt*), Arctic Tern (*at*), Common Tern (*ct*),
**Experiment 2** (*lt-at-al*): Least Tern (*lt*), Arctic Tern (*at*), Laysan Albatross (*al*),
**Experiment 3** (*lt-at-ca*): Least Tern (*lt*), Arctic Tern (*at*), Cardinal (*ca*).

Let $\Pr(lt\text{-}at \mid lt, at, ct)$ denote the probability of observing an edge between a Least Tern (*lt*) and an Arctic Tern (*at*) given that the 3 images in the query are Arctic Tern (*at*), Least Tern (*lt*), and Common Tern (*ct*). Similarly, we define $\Pr(lt\text{-}at \mid lt, at, al)$ and $\Pr(lt\text{-}at \mid lt, at, ca)$.

---

[1]When $m = 7, 8$, we set the number of queries to 9 and 8, respectively.

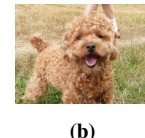 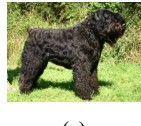 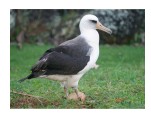 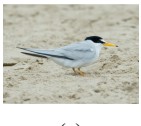 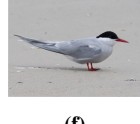 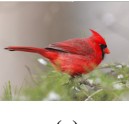 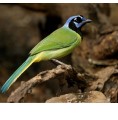 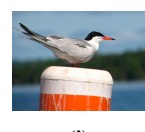

| (a) | (b) | (c) | (d) | (e) | (f) | (g) | (h) | (i) |

**Figure 5: (a) - (c) sample dogs (Norfolk Terrier, Toy Poodle, Bouvier des Flanders) of each species in the Dogs3 dataset. (d) - (i) sample birds (Albatross, Least Tern, Arctic Tern, Cardinal, Green Jay, Common Tern) of each species in the Birds5 and Birds5+ dataset.**

We aim to show that as the dissimilarity between the third image and the most similar pair of images in a query increase, the likelihood that the most similar pair being clustered together also increases. For example, $\Pr(lt\text{-}at \mid lt, at, al) \leq \Pr(lt\text{-}at \mid lt, at, ca)$, as a Cardinal is more dissimilar to the pair of Least Tern ($lt$) and Arctic Tern ($at$) than an Albatross ($al$) is. To obtain crowdworkers' perceptions regarding the similarity between the bird species in Birds5+ dataset, we query 20 crowdworkers for each pair of species. Each crowdworker answers 30 pairwise queries, with images randomly selected from the two pairs of species in Birds5+. Among the 30 pairs, 15 pairs involve images from different species, and the remaining 15 pairs involve images from the same species.

## 4 Results

### 4.1 Effect of Cognitive Overload in Multi-item Queries

We present the results obtained from the multi-item query experiment conducted using the drag-and-drop interface. For each experiment, we queried 300 unique crowdworkers. We defer the comparison between the two interfaces to Appendix C.5.

*Clustering Performance.* Figure 6 (a) reveals the performance of the clustering algorithms in terms of VI for Dogs3 and Birds5 . We note that for both datasets, VI reaches its minimum when $m$ is greater than or equal to 3. However, we begin to experience diminishing returns, especially in the Dogs3 dataset. Figure 6 (b) shows the worker edge error rate in the experiment, where a similar pattern of diminishing returns can be observed.

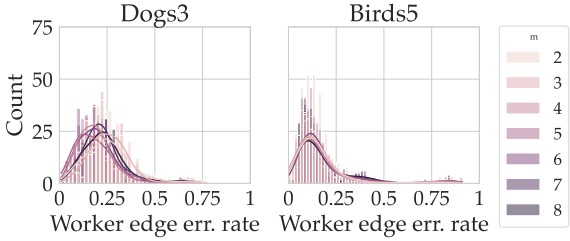

**Figure 7: Distribution of worker edge error rate for each number of items per query ($m$) on Dogs3 and Birds5 dataset.**

*Worker Edge Error Rate.* Figure 7 illustrates the distribution of the edge error rate for each crowdworker on Dogs3 and Birds5 dataset for each number of items per query $m$. We observe that the majority of the crowdworkers are better than random guessers: the worker edge error rate is less than 0.5.

*Edge Density between Clusters.* Figure 8 shows the empirical edge density within and between clusters on Dogs3 dataset. The values on the main diagonal represent the empirical probability of observing an edge given that the two images are from the same cluster. The other values on the $i^{\text{th}}$, $j^{\text{th}}$ column represent the empirical probability of observing an edge given that the two images are from clusters $i$ and $j$.

From these matrices, we observe that the probability of observing an edge between two different clusters when $2 < m < 5$ is smaller than when $m = 2$. This means that the adjacency matrix obtained from crowdworkers exhibits reduced ambiguity across different clusters. However, when $m \geq 5$, these probabilities start to increase, indicating a diminishing return. This aligns with the diminishing return we observed earlier. Similar pattern is shown on Birds5 dataset. We defer the presentation of the corresponding matrices to Appendix C.3.

*Summary.* Our results demonstrate that while increasing the query size theoretically improves crowdclustering performance, in practice, it provides no additional benefit to requesters when $m$ is larger than 4 or 5. We posit cognitive overload as an explanation. The task of comparing excessive images simultaneously (Figure 15 in the Appendix) overburdens crowdworkers.

### 4.2 Effect of Contextual bias

*Answer Distribution.* Table 1 presents the empirical observation probability matrix for the three experiments aimed at revealing the contextual bias of crowdworkers. Each column corresponds to an experiment, while each row is an answer pattern. The three numbers

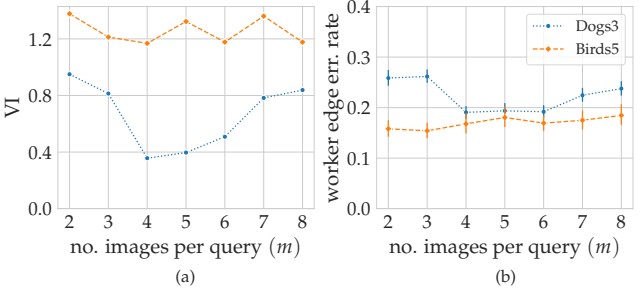

**Figure 6: Comparison of (a) time per query, (b) variation of information (VI), and (c) worker edge error rate between the Dogs3 and Birds5 datasets using the drag-and-drop interface, while varying the number of images per query.**

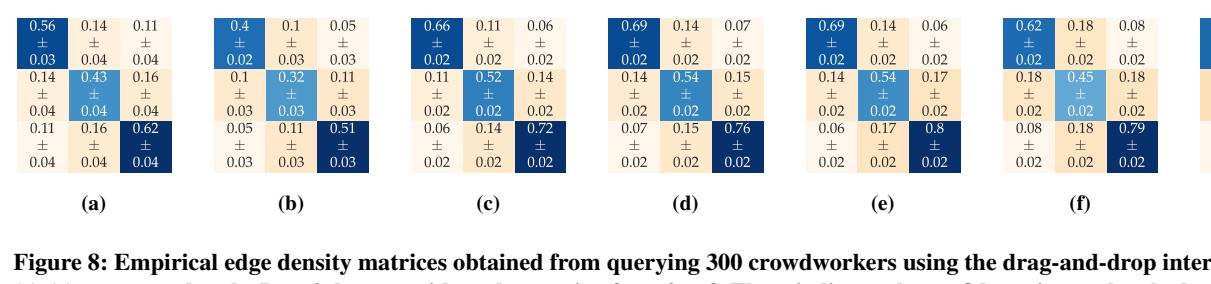

**(a)**    **(b)**    **(c)**    **(d)**    **(e)**    **(f)**    **(g)**

**Figure 8: Empirical edge density matrices obtained from querying 300 crowdworkers using the drag-and-drop interface. Matrices (a)-(g) correspond to the Dogs3 dataset, with $m$ also varying from 2 to 8. The ± indicates the confidence intervals calculated as described in Section 5.2.**

| Answer | lt-at-ct | lt-at-al | lt-at-ca |
|--------|----------|----------|----------|
| 1 1 1 | 0.152 ± 0.012 | 0.048 ± 0.007 | 0.030 ± 0.001 |
| 1 1 0 | 0.157 ± 0.012 | **0.443 ± 0.012** | **0.556 ± 0.016** |
| 1 0 1 | 0.192 ± 0.013 | 0.113 ± 0.010 | 0.062 ± 0.08 |
| 0 1 1 | **0.424 ± 0.016** | 0.112 ± 0.010 | 0.086 ± 0.009 |
| 0 1 2 | 0.075 ± 0.001 | 0.284 ± 0.015 | 0.266 ± 0.014 |

**Table 1: Empirical observation probability matrix for triangle query. Each column involves 1000 observations. The three numbers in the first column indicate the answer to the query. 0 1 2 means all three images are from different clusters; 1 0 1 means that the first and the third images are from the same cluster, etc.**

on the first column of each row represent an answer pattern. Where items at the positions where the numbers are the same are clustered together. For instance, 1 1 1 indicates all items are clustered together, while 1 1 0 means that the first two items form a cluster. Although the order of items varied between crowdworkers, we sort the items consistently for analysis. The sorted item order aligns with experiment names. For example, in experiment lt-at-ct, Least Tern (lt) is first, followed by Arctic Tern (at) and Common Tern (ct). Additionally, the sum of the first two rows for each column is the empirical estimate of the probability that lt and at get clustered together given the three items in the queries are lt, at, and ★ ($\Pr(lt\text{-}at \mid lt, at, \star)$), for $\star \in \{ct, al, ca\}$. For instance, the sum of the first two rows in the first column provides the empirical estimation for $\Pr(lt\text{-}at \mid lt, at, ct)$.

*Similarity Perception.* Table 2 reveals the empirical observation probability of pairs of images from different species in the Birds5+ dataset being clustered together. We treat these values as a surrogate for population perception of the similarity between each pair of species in the Birds5+ dataset. The higher the value, the more similar people consider the two species. From the table, we observe that Arctic Tern (at) and Common Tern (ct) are the most similar pair. This is what we expected, as shown in Figure 5, the difference between at and ct is subtle. Similarly, Least Tern (lt) and Cardinal (ca) are the most dissimilar pair, as lt and ca have completely different plumage.

With these values, we obtain the similarity between a species and a pair of species by averaging the empirical observation probability between the species and each in the pair. For example, the similarity between ca and the pair lt-at is $\frac{lt\text{-}ca + at\text{-}ca}{2} = \frac{0.060 + 0.093}{2} = 0.0765$.

*Context Biases Similarity Perception.* Figure 9 (a) reveals the relationship between the similarity of $\star \in \{ct, al, ca\}$ to the pair lt-at

and $\Pr(lt\text{-}at \mid lt, at, \star)$. It can be seen that as the similarity increases, the probability that lt and at are clustered together decreases. For example, as the third image changes from Cardinal (ca) to Common Tern (ct), the similarity of the third image to lt-at pair increases (since they all are Terns). Yet, the probability that lt and at get clustered together decreases.

We also perform bootstrapping on the observations from the three experiments by subsampling 75% of all observations 1000 times with replacement. Figure 9 (b) illustrates the distribution of $\Pr(lt\text{-}at \mid lt, at, \star)$ obtained by bootstrapping. This figure, together with Figure 9 (b), shows that when the third image is more similar to the pair lt, at, crowdworkers are more likely to differentiate the Terns, indicating that they focus on a different level in the hierarchy of details within different contexts.

Revisiting Table 1, the first column presents the empirical probability of each answer for **Experiment 1**: Least Tern (lt), Arctic Tern (at), Common Tern (ct). Since the at-ct pair is the most similar, the majority of crowdworkers group them together. However, when we see the second column, the most similar pair becomes lt-at. Therefore, the majority of people group lt-at together. When comparing the probability of grouping lt-at (sum of the first two rows) in the context of lt-at-ct, the value was much higher in the context of lt-at-al, and even higher in lt-at-ca. Figure 23 in the Appendix illustrates three sample triangle queries. When the third bird is much different from the first two birds, crowdworkers perceive on a higher level of similarity hierarchy, thus overlooking the minor differences between the two items. When the third bird is similar to the first two birds, crowdworkers consider similarity on a lower-level hierarchy, paying more attention to the details.

*Summary.* We argue that although technically, grouping lt-at together is incorrect, it is less erroneous than grouping at-al or at-ca together since they still differentiate Terns at a higher level from the other species. Therefore, it is important to reflect this phenomenon when modeling crowd noise.

## 5 Simulation Study

We use simulations to demonstrate that existing models cannot fully capture crowdworker errors, especially those due to contextual bias. The conditional block model (CBM) proposed in [45] has the potential to incorporate contextual bias in a crowdsourced clustering setting. In the following subsections, we define CBM, describe CBM simulation settings, and present CBM simulation results.

| pair type | lt-at | lt-ct | lt-al | lt-ca | at-ct | at-al | at-ca |
|---|---|---|---|---|---|---|---|
| probability | $0.410 \pm 0.028$ | $0.427 \pm 0.029$ | $0.120 \pm 0.019$ | $0.060 \pm 0.014$ | $\mathbf{0.767 \pm 0.024}$ | $0.210 \pm 0.024$ | $0.210 \pm 0.024$ |

**Table 2: Empirical observation probability of pairs of images from different species in Birds5+ being grouped together. Each type of pair is asked to 20 different people 15 times.**

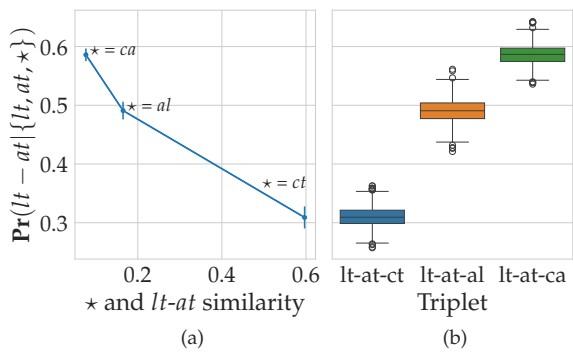

**Figure 9: (a) similarity between $\star$ and *lt-at* vs. $\Pr(lt\text{-}at|\{lt, at, \star\})$ for $\star \in \{ca, al, ct\}$ (b) distribution of $\Pr(lt\text{-}at|\{lt, at, \star\})$ from bootstrapping 75% of answers, for $\star \in \{ca, al, ct\}$.**

**Definition 5.1** (Conditional Block Model). A conditional block model (CBM) over a dataset of $n$ items that are partitioned by $K$ disjoint clusters and outliers $(C_1, C_2, \ldots, C_K)$ is a generative model parametrized by an edge density matrix $P \in [0, 1]^K$. Let $\text{cluster}(i) := k$ if $i \in C_k$. Then, given $m$ items, for each $(i, j)$ of all $\binom{m}{2}$ pairs of items, we draw an edge with probability $M_{\text{cluster}(i),\text{cluster}(j)}$. Note that not all possible generated configurations of these edges are admissible. In that case, we regenerate the configurations until an admissible one.

The above definition of CBM extends the CBM proposed in [45], which only accounts for three items per query, to multi-item queries. We simulate clustering results with different values of $m$ and different edge density matrices $P$.

**Remark 5.2.** Given $m$ items, there are $2^{\binom{m}{2}}$ possible ways of drawing edges among these items. However, not all configurations of these edge drawings are "reasonable". For example, in the case of $m = 3$, there are only five that are admissible (Figure 26a) out of the eight possibilities. This is due to the transitivity of "belonging to the same cluster". When item $i$ and item $j$ are put in the same cluster and item $j$ and item $k$ are in the same cluster, it is implied that item $j$ and $k$ are in the same cluster. Therefore, in CBM, when the outcome of drawing edges leads to an inadmissible configuration (Figure 26b), the CBM redraws the edges until an admissible configuration is obtained.

## 5.1 Simulation Settings

Let $r$ denote the proportion of edges observed (to all possible edges in the graph). For the first setting, we fix $r = 0.15$, the value used in [45]. In the second setting, we fix $r$ indirectly by fixing the total

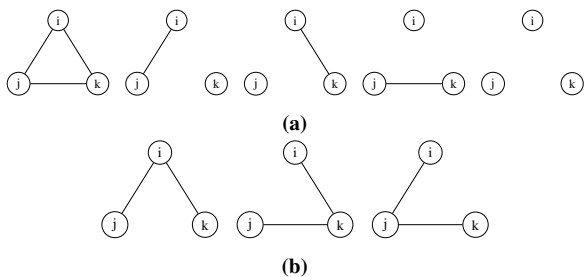

(a)

(b)

**Figure 10: Configurations for a three-item query that are (a) admissible and (b) inadmissible.**

number of edges explored at 9000, which is the number of edges explored in our crowdsourcing experiments. This way, we can compare the simulated results with our crowdsourcing experimental results.

*Setting 1: varying $p$.* We construct an edge density matrix whose main diagonal is $p$ and the off-diagonal is $q$. We vary $p$ from 0.55 to 1, with a step size of 0.05. We set $q = 0.25, K = 3,$ and $n = 300$. This is similar to the setting in [45]. For each $p$, we run the simulation ten times.

*Setting 2: using empirical edge density matrix.* We use the empirical edge density matrices obtained from our crowdsourcing experiments with the drag-and-drop interface and $m = 2$, as our edge density matrix $P$. We use $P$ to simulate the edge density for $m \geq 2$, with each $m$ 10 times. We run these simulations with settings similar to our experiments: $n = 473, K = 3$ for Dogs3 and $n = 342, K = 6$ [2] for Birds5. Lastly, we ensure the budget across different $m$ by fixing the total number of edges explored to 9000, and the number of queries for each $m$ is $2\frac{9000}{m}$.

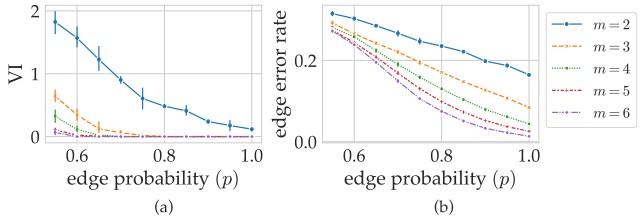

(a)

(b)

**Figure 11: CBM simulation results. The percentage of edges explored ($r$) is 0.15. (a) Variation of Information (VI) and (b) edge error rate at different (inter-cluster) edge probability ($p$) when the number of items to be clustered is 300, the number of clusters is 3, and $q = 0.25$, while varying edge density inside the clusters $p$.**

---

[2]We treat the outliers as one cluster.

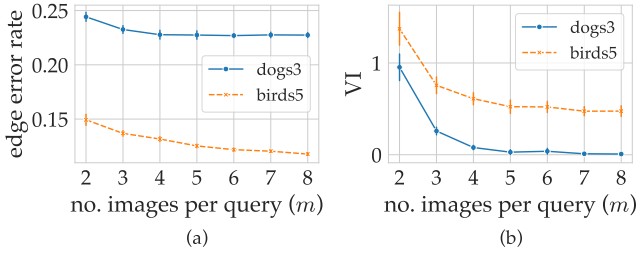

**Figure 12: (a) Variation of Information (VI) and (b) edge error rate at different number of images per query ($m$) when the edge density matrix comes from the crowdsourcing experiment. We fix the total number of edges explored to 9000.**

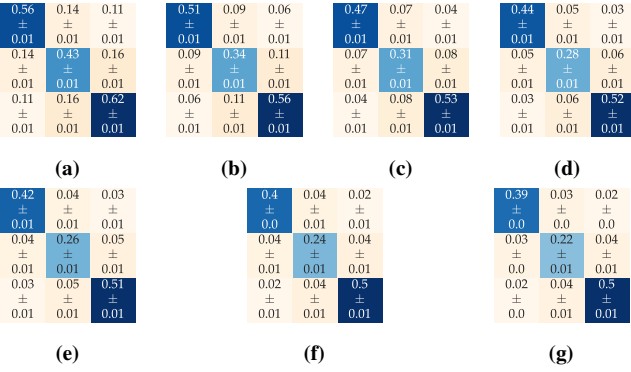

**Figure 13: Empirical edge density matrices obtained from simulation using the empirical edge density obtained from drag-and-drop interface when $m = 2$. Matrices (a)-(g) correspond to the Dogs3 dataset, with $m$ ranging from 2 to 8. The ± indicates the confidence intervals calculated as described in Section 5.2.**

## 5.2 Simulation Results

Figure 11 shows the variation of information (VI) and edge error rate as we vary the edge probability (density). It can be observed that as the number of items per query, $m$, increases, both VI and edge error rates decrease. However, the magnitude of improvement diminishes as $m$ increases, indicating a diminishing return effect.

Figure 12 illustrate the relationship between $m$ and VI as well as $m$ and the edge error rate. We observe that as $m$ increases, the magnitude of improvement of the two errors decreases, which can be considered as some diminishing return effect. However, there is still a difference between what we experimentally observed, where the diminishing return effect is more significant.

Figure 13 shows the edge density matrices obtained from the simulation by weighted-averaging each entry across the 10 edge density matrices. We use weighted-average here because the number of times an entry is observed for each edge density matrix is different. To help us compare these matrices to the ones we obtained from the experiment, we use Hoeffding's inequality to construct a concentration bound. Results regarding simulations on Birds5 and the concentration bound are deferred to the Appendix E.

## 6 Discussion

Our findings confirm prior literature by demonstrating the benefits of multi-item queries in crowdsourced clustering, but with diminishing returns beyond 4-5 items per query. This aligns with the established "magical number" concept in human information processing capacity [8, 28]. Additionally, our simulations, extending the Conditional Block Model (CBM) [45] to larger query sizes, reveal that the model does not fully capture this diminishing returns effect, suggesting a need for improved, more nuanced models for multi-item queries.

## 7 Conclusion

We examine the impact of cognitive overload and contextual bias of crowdworkers using simulations based on the conditional block model (CBM) and experiments conducted on AMT. Our simulations demonstrate that CBM does not fully explain the noise patterns observed in crowdsourcing. Moreover, in the experiments, we show that while there are advantages of having more items per query, these advantages tend to diminish after approximately 4 or 5 items per query. Furthermore, we discover that the noise in the answers varies depending on the specific items included in the query. The grouping of two items together relies not only on their relative similarity but also on the other item in the query.

Our results highlight the need for a more nuanced approach to modeling noise in crowdsourcing tasks, as current models fail to capture the underlying structure within the noise, which is crucial in practical applications. In the future, we will provide a more theoretical analysis of the guaranteed recovery of the actual adjacency matrix.

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

## A User Interface

We develop our interface using React.js [27], which compiles into a static website. We host our website on AWS S3 [4] and use AWS Lambda [3] to bridge the connection between the static website and our database hosted on MongoDB Atlas [30]. When a crowdworker accepts our HIT, he or she is given a link to our website. To enhance the experiment's efficiency, each crowdworker is required to complete more than one query. When a crowdworker completes all queries, their answers are saved to our dataset. Upon completion, the website generates a unique identifier for the crowdworker. The crowdworker must input this identifier into a text box in AMT before submitting this task on AMT. We use this unique identifier to make sure that we only analyze answers submitted by crowdworkers who submit this task [3].

## B Crowdsourcing Study

### B.1 Tutorial Stage

In the tutorial stage of the drag-and-drop interface, a series of prompts teach a crowdworker how to manipulate the interface. At the end of the tutorial stage, the crowdworker must accurately group a set of $m$ items. Note that the set of items shown in the tutorial stage is fixed for all crowdworkers; and they are manually picked by us so that it is very easy for crowdworkers to give a correct grouping.

### B.2 Practice Stage

After the tutorial stage, the crowdworkers need to complete 3 easy queries. Similarly to the tutorial stage, the items in the practice stage are fixed across crowdworkers and are maually picked by us. For each practice query, a crowdworker has 3 chances to give a correct grouping. If an incorrect grouping is given, our system will prompt the crowdworker that a mistake is made and they need to redo the practice query.

---

[3]We host our website on the Internet. Technically, everyone can access the website if they know the URL to it. Although this is unlikely, we still add this step to ensure that we only analyze those crowdworkers who accepted the task. We do so by checking if the unique ID of an answer exists in AMT.

# Instructions:

- Thank you for your interest!
- You will be shown 15 questions of images with birds in them.
- For each question, you will see 15 radio buttons.
- For the 4 images shown to you,
  - If you think birds in the 4 images are of the same species, then click on `All are Same Species`
  - If you think only birds in Image 1, 2, and 3 are from the same species, then click on `ONLY 1, 2, and 3 are Same Species`
  - If you think Image 1 and 2 are of one species and Image 3 and 4 are of a different species, then click on `1 and 2 are Same Species; 3 and 4 are Same Species`
  - If you think Image 1 and 2 are of one species and Image 3 is of its own species, and Image 4 is of its own species, then click on `1 and 2 are Same Species; 3 is a different Species; 4 is a different Species`
  - If you think all are different species, then click on `NONE, all are different species`
- You need to answer all the questions.
- Of the 15, there are 3 (random out of 15) GOLD STANDARD questions. You need to get at least 2 of them correct to get the answers accepted.

## Question 1 / 15

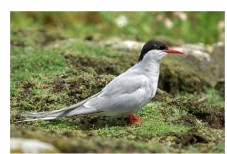 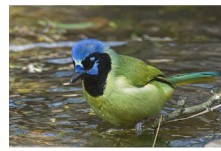 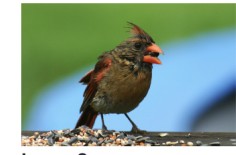 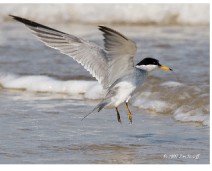

**Image 1**  **Image 2**  **Image 3**  **Image 4**

○ All are Same Species
○ ONLY 1, 2, and 3 are Same Species
○ ONLY 1, 2, and 4 are Same Species
○ ONLY 1, 3, and 4 are Same Species
○ ONLY 2, 3, and 4 are Same Species
○ 1 and 2 are Same Species; 3 and 4 are Same Species
○ 1 and 3 are Same Species; 2 and 4 are Same Species
○ 1 and 4 are Same Species; 2 and 3 are Same Species
○ 1 and 2 are Same Species; 3 is a different Species; 4 is a different Species
○ 1 and 3 are Same Species; 2 is a different Species; 4 is a different Species
○ 1 and 4 are Same Species; 2 is a different Species; 3 is a different Species
○ 2 and 3 are Same Species; 1 is a different Species; 4 is a different Species
○ 2 and 4 are Same Species; 1 is a different Species; 3 is a different Species
○ 3 and 4 are Same Species; 1 is a different Species; 2 is a different Species
○ NONE, all are different species

Next

**Figure 14: Samples of the radio interface deployed on Amazon Mechanical Turk with $m = 4$ and Dogs3 dataset. Instructions are always shown on top of the page. Crowdworkers need to click on the Next button to proceed to the next query. As we can see, when $m = 4$, the number of radio buttons is 15. These many radio buttons may induce cognitive overload.**

## C Additional Results

### C.1 Distribution of time per query

Figure 16 illustrates the distribution of time per query on Dogs3 datasets. Figure 17 illustrate the distribution of time per query for datasets. We can observe that the distributions shift to the right when $m$ increases in the two datasets.

### C.2 Distribution of worker edge error rate

Figure 18 and 19 illustrates the distribution of the edge error rate for each crowdworker on Dogs3 and Birds5 dataset for each number of items per query $m$. We observe that the majority of the crowdworkers are better than random guessers: the worker edge error rate is less than 0.5.

### C.3 Empirical edge density

Figure 20 and 21 show the empirical edge density both within and between clusters. The values on the main diagonal represent the empirical probability of observing an edge given that the two images are from the same cluster. The other values on the $i$-th row, $j$-th column represent the empirical probability of observing an edge given that the two images are from clusters $i$ and $j$.

From these matrices, we observe that as $m$ increases, the probability of observing an edge between two different clusters decreases. This means that the adjacency matrix obtained from crowdworkers exhibits reduced ambiguity across different clusters. Although the edge probability between images from the same clusters also decreases as $m$ increases, the benefits of reducing the ambiguity across different clusters outweigh this. This is because exploring more edges, when $m$ is larger, with some level of uncertainty can be more beneficial than gathering a smaller number of high-quality, precise edges [46]. However, when $m \geq 4$, the ambiguity between difference clusters reemerges. This aligns with the diminishing return we observed earlier.

### C.4 Cost

Figure 25a illustrates the amount of time crowdworkers spend on each query and exhibits a roughly linear relationship with the number of images per query, for both Dogs3 and Birds5 datasets. Since we fixed the total number of images each crowdworker could see (except for $m = 7, 8$), we compensated crowdworkers the same amount of money across different $m$. Figures 16 and 17 illustrate the

**Figure 15: Sample of the drag-and-drop interface deployed on Amazon Mechanical Turk with $m = 2$ and Dogs3 dataset. Instructions are always shown on the top of the page. Crowdworkers can drag the images to an existing cluster or form a new cluster.**

distribution of time per query for the two datasets. We can observe that the distribution moves to the right when $m$ increases in the two datasets.

## C.5    Other observations

*Our Radio vs. Prior Radio interface.* Using the Dogs3 dataset, we were able to replicate the prior results from [45], which demonstrated better clustering performance with triangle queries compared to edge queries. Additionally, we also observed diminishing returns in VI with increasing $n_p$ (the number of items per query). Specifically, when moving from 2 to 4 images per query, the VI worsened. This can be attributed to the mental exhaustion caused by the excessive options presented in the radio button interface (as shown in Figure 14), which can lead to misclicks and errors. We again note that for $m \geq 5$, radio button interface is too cumbersome. Hence, we design the drag-and-drop interface (Section 3.5).

Figure 22 illustrates the empirical edge density within and across clusters. We observe a similar pattern to the results reported in [45] and in Section 4.1. As $m$ increases from 2 to 3, the adjacency matrix obtained from crowdworkers exhibits reduced ambiguity across different clusters. However, at $m = 4$, the ambiguity strikes back. This finding aligns with the diminishing returns observed in the table.

*Radio vs. Drag-and-drop interface.* Tables 3 and 4 compare the crowdclustering outcomes between the radio and the drag-and-drop interfaces on the Dogs3 and Birds5 datasets, respectively. For both datasets, the variation of information (VI) is lower with the drag-and-drop interface compared to the radio interface, indicating a significant impact of interface design on noise levels and subsequent denoising performance. However, it should be noted that the drag-and-drop queries take longer to complete, which could increase the total collection time and expenses

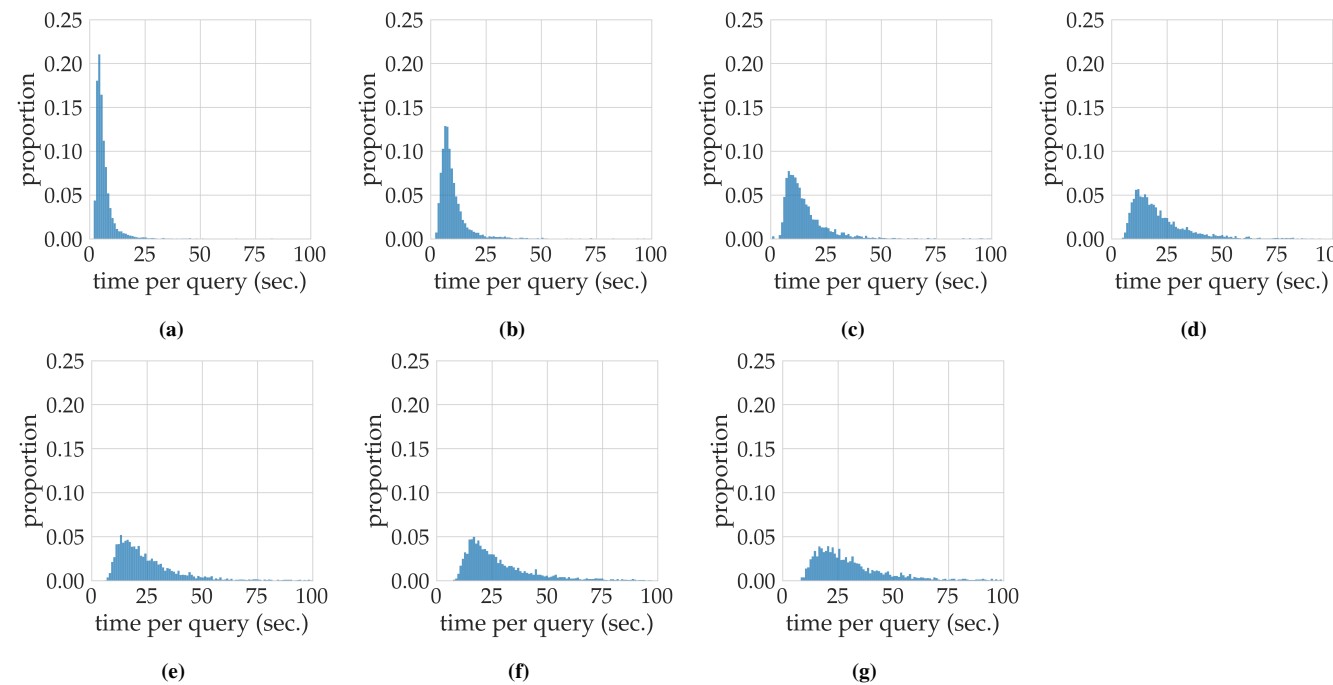

**Figure 16: Distribution of time per query for Dogs3 dataset. (a)-(g) correspond to the number of items per query $m$ from 2-8.**

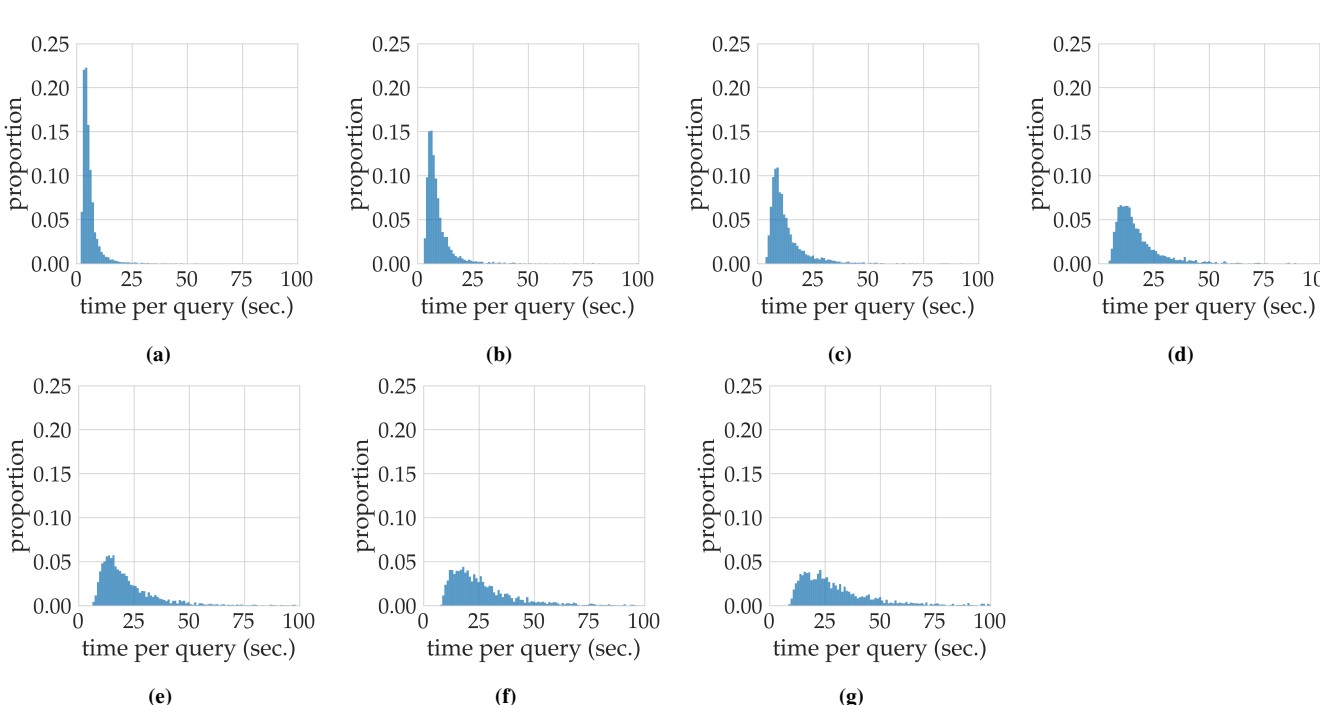

**Figure 17: Distribution of time per query for Birds5 dataset. (a)-(g) correspond to the number of items per query $m$ from 2-8.**

for requesters.

## C.6 Hierarchy of Birds

Figure 23 illustrates the three triangle queries. When the third bird is much different from the first two birds, crowdworkers perceive on a higher level of similarity hierarchy, thus overlooking the minor differences between the two

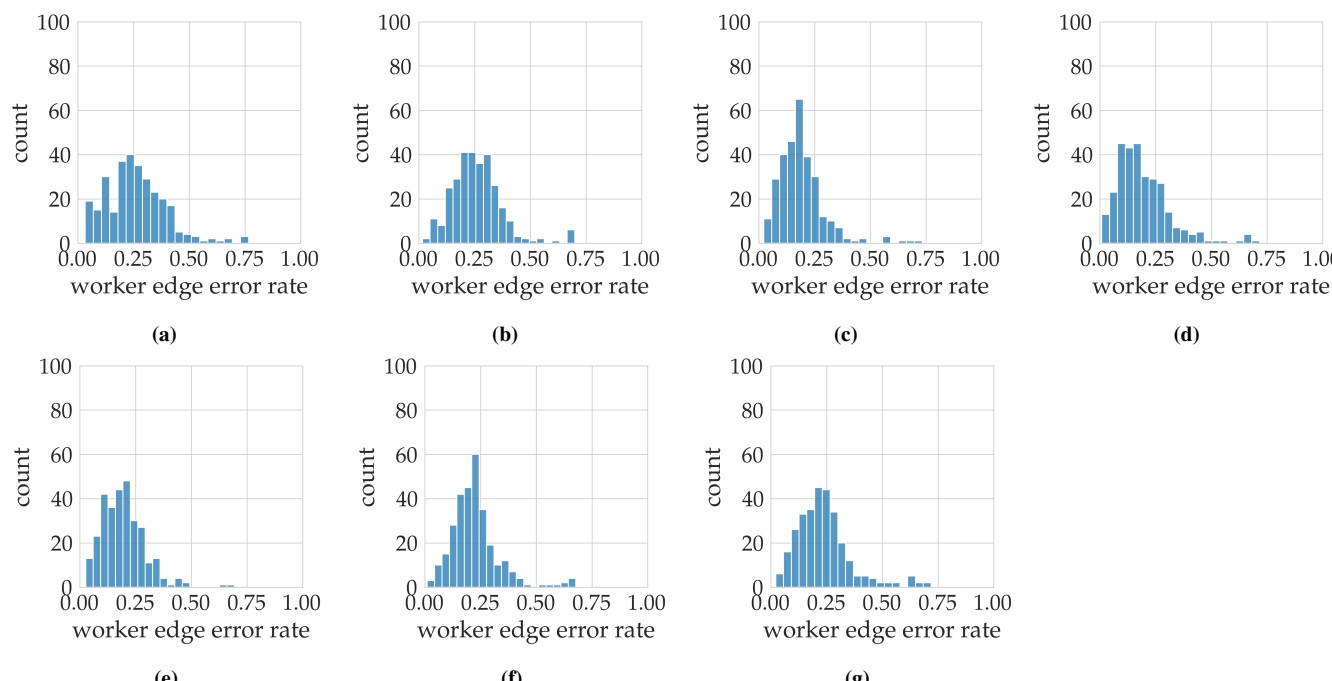

**Figure 18: Distribution of worker edge error rate for Dogs3 dataset. (a)-(g) correspond to the number of items per query $m$ from 2-8.**

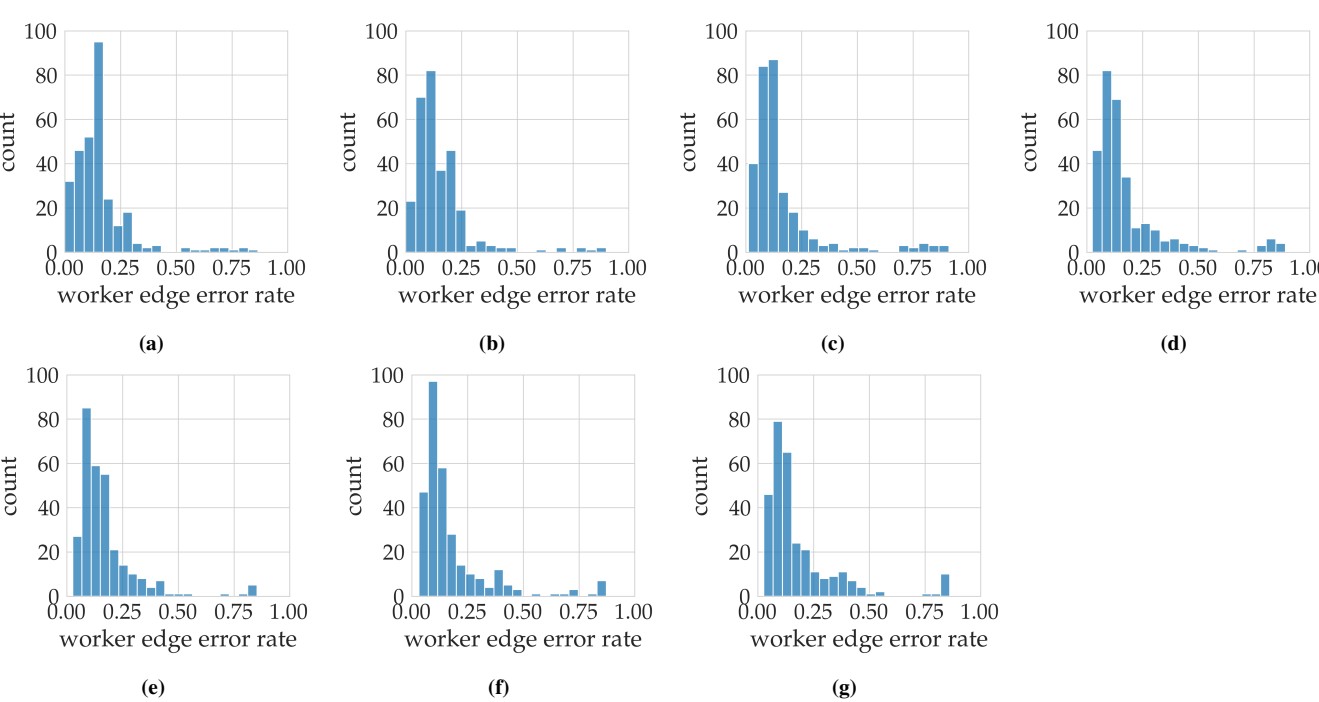

**Figure 19: Distribution of worker edge error rate for Birds5 dataset. (a)-(g) correspond to the number of items per query $m$ from 2-8.**

items. When the third bird is similar to the first two birds, crowdworkers consider similarity on a lower-level hierarchy, paying more attention to the details.

## D  Conditional Block Model

**Definition D.1** (Stochastic Block Model).  A stochastic block model (SBM) over a dataset of $n$ items that are partitioned by $K$ disjoint clusters and outliers is a generative model parametrized by $0 < p, q < 1$. Given a pair of items $i, j$,

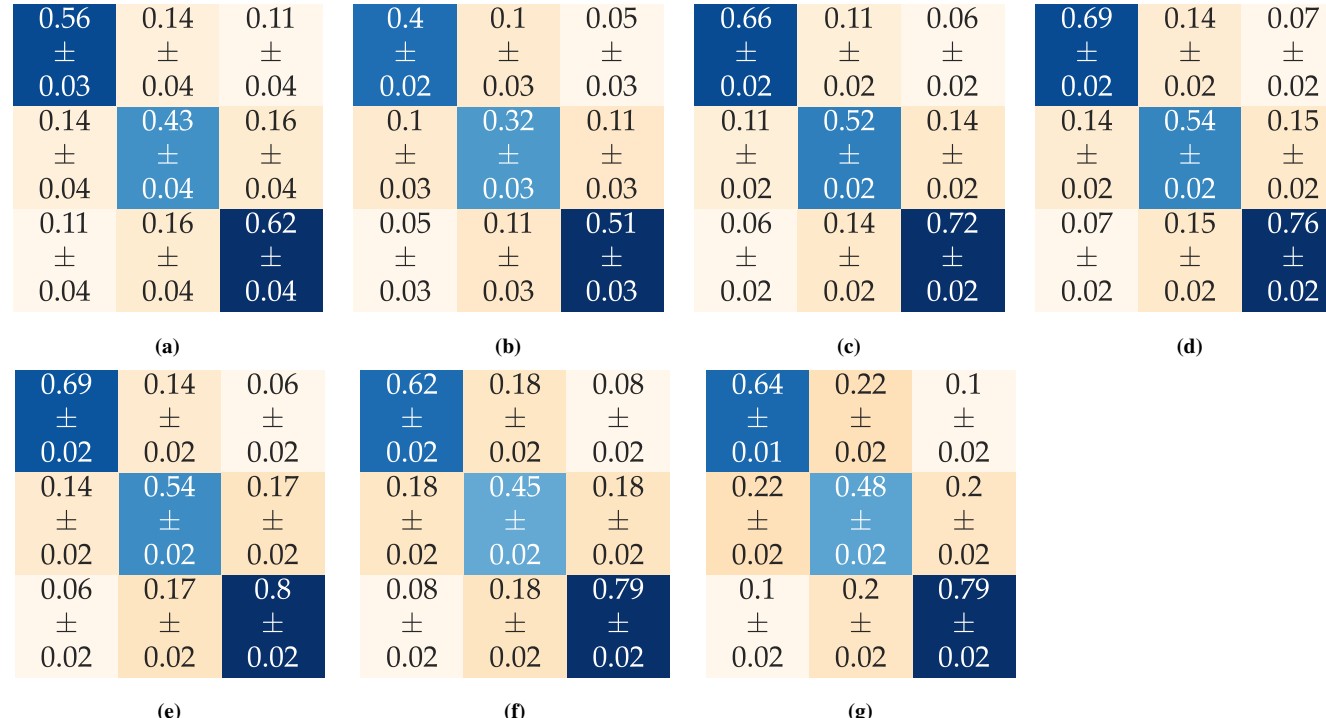

**Figure 20: Empirical edge density matrices obtained from querying 300 crowdworkers using the drag-and-drop interface. Matrices (a)-(g) correspond to the Dogs3 dataset, with $m$ also varying from 2 to 8. The $\pm$ indicates the confidence intervals calculated as described in Section 5.2.**

| m | radio interface | | | | drag-and-drop interface | | | |
|---|---|---|---|---|---|---|---|---|
| | no. unique edges | VI | edge error rate | time per query (sec.) | no. unique edges | VI | edge error rate | time per query (sec.) |
| 2 | 8,663 | 1.576 | **0.314 ± 0.005** | **5.356 ± 11.567** | 8,637 | 0.950 | 0.258 ± 0.005 | **6.952 ± 8.895** |
| 3 | 16,619 | **1.520** | 0.321 ± 0.003 | 7.207 ± 14.054 | 16,632 | 0.814 | 0.262 ± 0.003 | 10.828 ± 13.319 |
| 4 | **24,044** | 1.754 | 0.366 ± 0.003 | 10.656 ± 20.475 | **23,968** | 0.357 | **0.191 ± 0.002** | 16.626 ± 18.445 |

**Table 3: Number of edges explored, VI for the clustering outcome, edge error rate, and mean completion time of a single question among all crowdworkers who participate in the experiment using the radio interface and drag-and-drop interface on Dogs3 Dataset.**

| m | radio interface | | | | drag-and-drop interface | | | |
|---|---|---|---|---|---|---|---|---|
| | no. unique edges | VI | edge error rate | time per query (sec.) | no. unique edges | VI | edge error rate | time per query (sec.) |
| 2 | 8,393 | 2.449 | 0.264 ± 0.003 | **4.779 ± 10.550** | 8,341 | 1.378 | 0.157 ± 0.004 | **6.694 ± 13.259** |
| 3 | 15,513 | **2.233** | **0.232 ± 0.003** | 7.024 ± 14.937 | 15,542 | 1.213 | **0.154 ± 0.003** | 10.140 ± 17.239 |
| 4 | **21,633** | 2.698 | 0.265 ± 0.005 | 9.914 ± 26.283 | **21,551** | 1.169 | 0.167 ± 0.002 | 14.667 ± 20.090 |

**Table 4: Number of edges explored, VI for the clustering outcome, edge error rate, and mean completion time of a single question among all crowdworkers who participate in the experiment using the radio interface and drag-and-drop interface on Birds5 Dataset.**

we draw an edge between them given that they are from the same cluster, with probability $p$; we draw an edge between them given that they are not from the same cluster, with probability $q$.

**Definition D.2** (three-item Conditional Block Model [45]). A three-item conditional block model (CBM) over a dataset of $n$ items that are partitioned by $K$ disjoint clusters and outliers $(C_1, C_2, \ldots, C_K)$ is a generative model parametrized by $0 < p, q < 1$. Given 3 items, there could be 8 edge configurations (see 26), among which 5 are admissible. We first draw edges between each pair of the 3

items following SBM. If the resulting configuration is inadmissible, we regenerate configurations until an admissilbe one.

We extend the three-item conditional block model proposed in [45] for multi-item query, and we use a matrix $P$ to parametrize the model to account for different inter-cluster edge densities between different clusters.

**Definition D.3** (Conditional Block Model). A conditional block model (CBM) over a dataset of $n$ items that are partitioned by $K$ disjoint clusters and outliers $(C_1, C_2, \ldots, C_K)$ is a generative model parametrized by an edge density matrix

**(a)**

| | | | | | |
|---|---|---|---|---|---|
| 0.55 ± 0.08 | 0.14 ± 0.08 | 0.09 ± 0.08 | 0.05 ± 0.08 | 0.04 ± 0.08 | 0.06 ± 0.09 |
| 0.14 ± 0.08 | 0.51 ± 0.08 | 0.39 ± 0.08 | 0.04 ± 0.08 | 0.06 ± 0.08 | 0.06 ± 0.09 |
| 0.09 ± 0.08 | 0.39 ± 0.08 | 0.59 ± 0.08 | 0.05 ± 0.09 | 0.05 ± 0.08 | 0.07 ± 0.09 |
| 0.05 ± 0.08 | 0.04 ± 0.08 | 0.05 ± 0.09 | 0.67 ± 0.08 | 0.06 ± 0.09 | 0.09 ± 0.1 |
| 0.04 ± 0.08 | 0.06 ± 0.08 | 0.05 ± 0.08 | 0.06 ± 0.09 | 0.74 ± 0.08 | 0.09 ± 0.09 |
| 0.06 ± 0.09 | 0.06 ± 0.09 | 0.07 ± 0.09 | 0.09 ± 0.1 | 0.09 ± 0.09 | 0.27 ± 0.11 |

**(b)**

| | | | | | |
|---|---|---|---|---|---|
| 0.52 ± 0.06 | 0.15 ± 0.06 | 0.12 ± 0.06 | 0.04 ± 0.06 | 0.05 ± 0.06 | 0.06 ± 0.07 |
| 0.15 ± 0.06 | 0.51 ± 0.06 | 0.33 ± 0.06 | 0.05 ± 0.06 | 0.05 ± 0.06 | 0.09 ± 0.07 |
| 0.12 ± 0.06 | 0.33 ± 0.06 | 0.6 ± 0.06 | 0.05 ± 0.06 | 0.06 ± 0.06 | 0.08 ± 0.07 |
| 0.04 ± 0.06 | 0.05 ± 0.06 | 0.05 ± 0.06 | 0.67 ± 0.06 | 0.07 ± 0.06 | 0.07 ± 0.07 |
| 0.05 ± 0.06 | 0.05 ± 0.06 | 0.06 ± 0.06 | 0.07 ± 0.06 | 0.82 ± 0.06 | 0.11 ± 0.07 |
| 0.06 ± 0.07 | 0.09 ± 0.07 | 0.08 ± 0.07 | 0.07 ± 0.07 | 0.11 ± 0.07 | 0.2 ± 0.08 |

**(c)**

| | | | | | |
|---|---|---|---|---|---|
| 0.56 ± 0.05 | 0.17 ± 0.05 | 0.15 ± 0.05 | 0.06 ± 0.05 | 0.07 ± 0.05 | 0.11 ± 0.06 |
| 0.17 ± 0.05 | 0.55 ± 0.05 | 0.37 ± 0.05 | 0.08 ± 0.05 | 0.08 ± 0.05 | 0.1 ± 0.06 |
| 0.15 ± 0.05 | 0.37 ± 0.05 | 0.66 ± 0.05 | 0.08 ± 0.05 | 0.08 ± 0.05 | 0.09 ± 0.06 |
| 0.06 ± 0.05 | 0.08 ± 0.05 | 0.08 ± 0.05 | 0.72 ± 0.05 | 0.1 ± 0.05 | 0.09 ± 0.06 |
| 0.07 ± 0.05 | 0.08 ± 0.05 | 0.08 ± 0.05 | 0.1 ± 0.05 | 0.84 ± 0.05 | 0.12 ± 0.06 |
| 0.11 ± 0.06 | 0.1 ± 0.06 | 0.09 ± 0.06 | 0.09 ± 0.06 | 0.12 ± 0.06 | 0.18 ± 0.06 |

**(d)**

| | | | | | |
|---|---|---|---|---|---|
| 0.54 ± 0.04 | 0.17 ± 0.04 | 0.16 ± 0.04 | 0.07 ± 0.05 | 0.07 ± 0.05 | 0.12 ± 0.05 |
| 0.17 ± 0.04 | 0.55 ± 0.04 | 0.36 ± 0.04 | 0.08 ± 0.04 | 0.09 ± 0.05 | 0.12 ± 0.05 |
| 0.16 ± 0.04 | 0.36 ± 0.04 | 0.62 ± 0.05 | 0.09 ± 0.05 | 0.09 ± 0.05 | 0.12 ± 0.05 |
| 0.07 ± 0.05 | 0.08 ± 0.04 | 0.09 ± 0.05 | 0.71 ± 0.05 | 0.12 ± 0.05 | 0.13 ± 0.05 |
| 0.07 ± 0.05 | 0.09 ± 0.05 | 0.09 ± 0.05 | 0.12 ± 0.05 | 0.83 ± 0.05 | 0.14 ± 0.05 |
| 0.12 ± 0.05 | 0.12 ± 0.05 | 0.12 ± 0.05 | 0.13 ± 0.05 | 0.14 ± 0.05 | 0.19 ± 0.06 |

**(e)**

| | | | | | |
|---|---|---|---|---|---|
| 0.51 ± 0.04 | 0.15 ± 0.04 | 0.16 ± 0.04 | 0.04 ± 0.04 | 0.05 ± 0.04 | 0.08 ± 0.05 |
| 0.15 ± 0.04 | 0.55 ± 0.04 | 0.38 ± 0.04 | 0.05 ± 0.04 | 0.08 ± 0.04 | 0.11 ± 0.05 |
| 0.16 ± 0.04 | 0.38 ± 0.04 | 0.57 ± 0.04 | 0.05 ± 0.04 | 0.07 ± 0.04 | 0.09 ± 0.05 |
| 0.04 ± 0.04 | 0.05 ± 0.04 | 0.05 ± 0.04 | 0.71 ± 0.04 | 0.11 ± 0.04 | 0.12 ± 0.05 |
| 0.05 ± 0.04 | 0.08 ± 0.04 | 0.07 ± 0.04 | 0.11 ± 0.04 | 0.81 ± 0.04 | 0.14 ± 0.05 |
| 0.08 ± 0.05 | 0.11 ± 0.05 | 0.09 ± 0.05 | 0.12 ± 0.05 | 0.14 ± 0.05 | 0.21 ± 0.05 |

**(f)**

| | | | | | |
|---|---|---|---|---|---|
| 0.56 ± 0.04 | 0.17 ± 0.04 | 0.17 ± 0.04 | 0.06 ± 0.04 | 0.07 ± 0.04 | 0.1 ± 0.04 |
| 0.17 ± 0.04 | 0.58 ± 0.04 | 0.43 ± 0.04 | 0.08 ± 0.04 | 0.09 ± 0.04 | 0.14 ± 0.04 |
| 0.17 ± 0.04 | 0.43 ± 0.04 | 0.68 ± 0.04 | 0.08 ± 0.04 | 0.09 ± 0.04 | 0.12 ± 0.04 |
| 0.06 ± 0.04 | 0.08 ± 0.04 | 0.08 ± 0.04 | 0.74 ± 0.04 | 0.13 ± 0.04 | 0.14 ± 0.04 |
| 0.07 ± 0.04 | 0.09 ± 0.04 | 0.09 ± 0.04 | 0.13 ± 0.04 | 0.91 ± 0.04 | 0.15 ± 0.04 |
| 0.1 ± 0.04 | 0.14 ± 0.04 | 0.12 ± 0.04 | 0.14 ± 0.04 | 0.15 ± 0.04 | 0.19 ± 0.05 |

**(g)**

| | | | | | |
|---|---|---|---|---|---|
| 0.6 ± 0.04 | 0.21 ± 0.04 | 0.21 ± 0.04 | 0.07 ± 0.04 | 0.08 ± 0.04 | 0.11 ± 0.04 |
| 0.21 ± 0.04 | 0.58 ± 0.04 | 0.44 ± 0.04 | 0.1 ± 0.04 | 0.11 ± 0.04 | 0.13 ± 0.04 |
| 0.21 ± 0.04 | 0.44 ± 0.04 | 0.69 ± 0.04 | 0.08 ± 0.04 | 0.09 ± 0.04 | 0.12 ± 0.04 |
| 0.07 ± 0.04 | 0.1 ± 0.04 | 0.08 ± 0.04 | 0.75 ± 0.04 | 0.14 ± 0.04 | 0.14 ± 0.04 |
| 0.08 ± 0.04 | 0.11 ± 0.04 | 0.09 ± 0.04 | 0.14 ± 0.04 | 0.86 ± 0.04 | 0.17 ± 0.04 |
| 0.11 ± 0.04 | 0.13 ± 0.04 | 0.12 ± 0.04 | 0.14 ± 0.04 | 0.17 ± 0.05 | 0.2 ± 0.05 |

**Figure 21: Empirical edge density matrices obtained from querying 300 crowdworkers using the drag-and-drop interface. Matrices (a)-(g) correspond to the Birds5 dataset, with $m$ ranging from 2 to 8. The ± indicates the confidence intervals calculated as described in Section 5.2.**

$P \in [0, 1]^K$. Let cluster$(i) := k$ if $i \in C_k$. Then, given $m$ items, for each $(i, j)$ of all $\binom{m}{2}$ pairs of items, we draw an edge with probability $M_{\text{cluster}(i),\text{cluster}(j)}$. Note that not all possible generated configurations of these edges are admissible. In that case, we regenerate configurations until an admissible one.

Given $m$ items, there are $2^{\binom{m}{2}}$ possible ways of drawing edges among these items. However, not all configurations of these edge drawings are "reasonable". For example, in the case of $m = 3$, there are only five that are admissible (Figure 26a) out of the eight possibilities. This is due to the transitivity of "belonging to the same cluster". When item $i$ and item $j$ are put in the same cluster and item

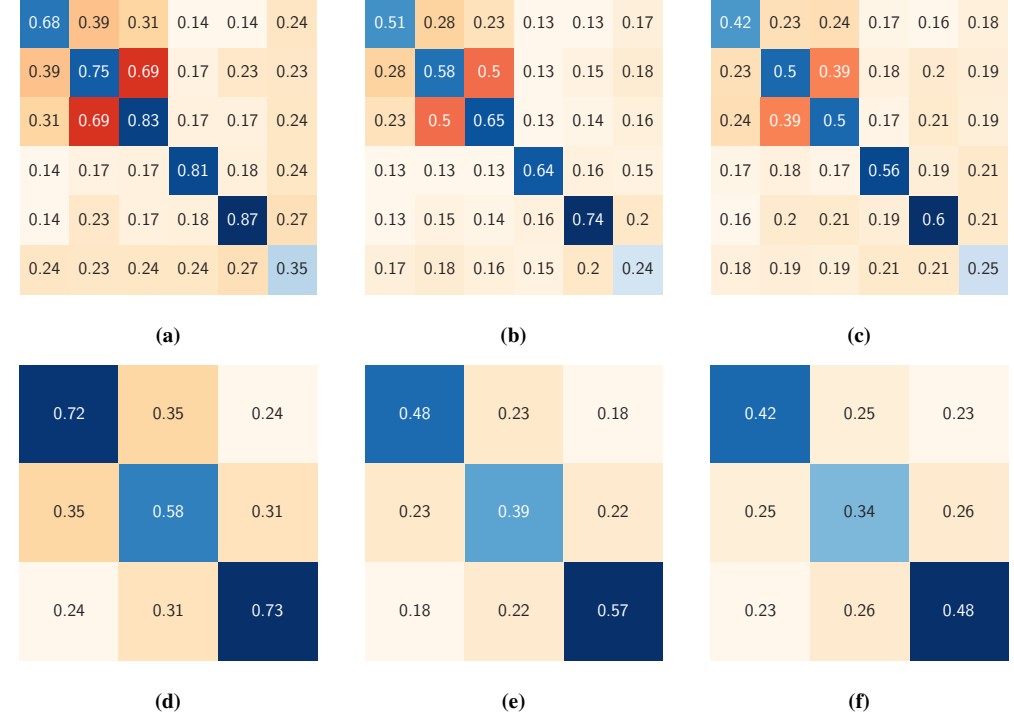

**Figure 22: Empirical edge density matrices obtained from querying 300 crowdworkers using the radio interface. Matrices (a)-(c) correspond to the Birds5 dataset, with $m$ ranging from 2 to 8. Matrices (d)-(f) correspond to the Dogs3 dataset, with $m$ also varying from 2 to 8.**

$j$ and item $k$ are in the same cluster, it is implied that item $j$ and $k$ are in the same cluster. Therefore, in CBM, when the outcome of drawing edges leads to an inadmissible configuration (Figure 26b), the CBM redraws the edges until an admissible configuration is obtained.

The edge density matrix $P$ that parametrizes the model represents the inter and intra cluster density. The entries on the main-diagonal represent the intra-cluster density whereas the off-diagonal entries represent the inter-cluster density. Note that the edge density matrix is a generalization of the $p$ and $q$ in the classic SBM. When the main-diagonal and off-diagonal entries are fixed to $p$ and $q$, we draw edges exactly like SBM.

The above definition of CBM extends the CBM proposed in [45], which only accounts for three items per query, to multi-item queries. We simulate clustering results with different values of $m$ and different edge density matrices $P$.

## E   Simulation Concentration Bounds

Let $P \in [0,1]^{d \times d}$ denote an edge density matrix, and let $O \in \mathbb{N}^{d \times d}$ denote an edge observation matrix, where $O_{ij}$ denotes the number of times this entry is observed. We use Hoeffding's inequality [15, 51], which provides the concentration bound $P_{ij} \pm \epsilon$ such that the expected $P_{ij}$ falls out of the interval defined by the bound with probability at most $\delta'$:

$$\mathbb{P}(|P_{ij} - \mathbb{E}[P_{ij}]| \geq \epsilon) \leq 2\exp(-2 \cdot O_{ij} \cdot \epsilon^2) = \delta'. \quad (1)$$

Figure 27 and 28 show the edge density matrices obtained from the simulation by weighted-averaging each entry across the 10 edge density matrices from the simulations. We use weighted-average here because the number of times an entry is observed for each edge density matrix is different. To help us compare these matrices to the ones we obtained from the experiment, we use Hoeffding's inequality to construct a concentration bound.

By definition, $P$ is a symmetric matrix. Therefore, there are $\frac{d^2-d}{2} + d$ unique entries. We apply a union-bound correction to compare all entries at the same time:

$$\mathbb{P}\left(\bigcup_{1 \leq i \leq j \leq d} |P_{ij} - \mathbb{E}[P_{ij}]| \geq \epsilon\right) \quad (2)$$

$$\leq \sum_{1 \leq i \leq j \leq d} 2\exp\left(-2 \cdot O_{ij} \cdot \epsilon^2\right) \quad (3)$$

$$= (d^2 + d)\exp(-2 \cdot O_{ij} \cdot \epsilon^2) = \delta. \quad (4)$$

Therefore, with union bound correction, for each edge density matrix being compared, we can have a confidence intervals for all the unique entries that are simultaneously valid with probability at least $1 - \delta$.

Figures 29 and 30 illustrate the concentration bound ($\delta = 0.05$) for Dogs3 and Birds5 dataset, while we vary $m$ from 2 to 8. Having $\delta = 0.05$ means that the probability that the expected entry value falls out of its corresponding interval is at most 0.05. Figures 31a and 31b show the number of mismatches between the two bounds for Dogs3 and Birds5 dataset, as a function of $m$. We observe that as $m$ increases, the number of entries in which the two bounds do not overlap increases. This means that there is a huge difference between the edge density predicted by CBM and the one we observed from the experiment once $m$ is greater than 2. Therefore, there must exist other factors influencing the 'noises' that the model failed to capture. And one of the factors could be the contextual bias we described in the previous section.

## F   Extended Related Works

### F.1   Direct Labeling Query

Many works on theoretical understanding of crowdsourcing focus on labeling tasks, where crowdworkers are asked to label items directly [9, 13, 17–19, 24, 34, 40 ? ? ]. Karger et al. and Karger et al. [17, 18] adopted the "spammer-hammer model", which treats workers as a mixture of "spammers", who randomly answer the questions, and "hammers", who answer correctly. Mazumdar and Pal and Pang et al. [24, 34] treated each query as a function that takes $n$ items as input and outputs 0 or 1. They utilized methods from information theory and coding theory to reconstruct the labeling from the answers to the queries. Mazumdar and Pal [24]

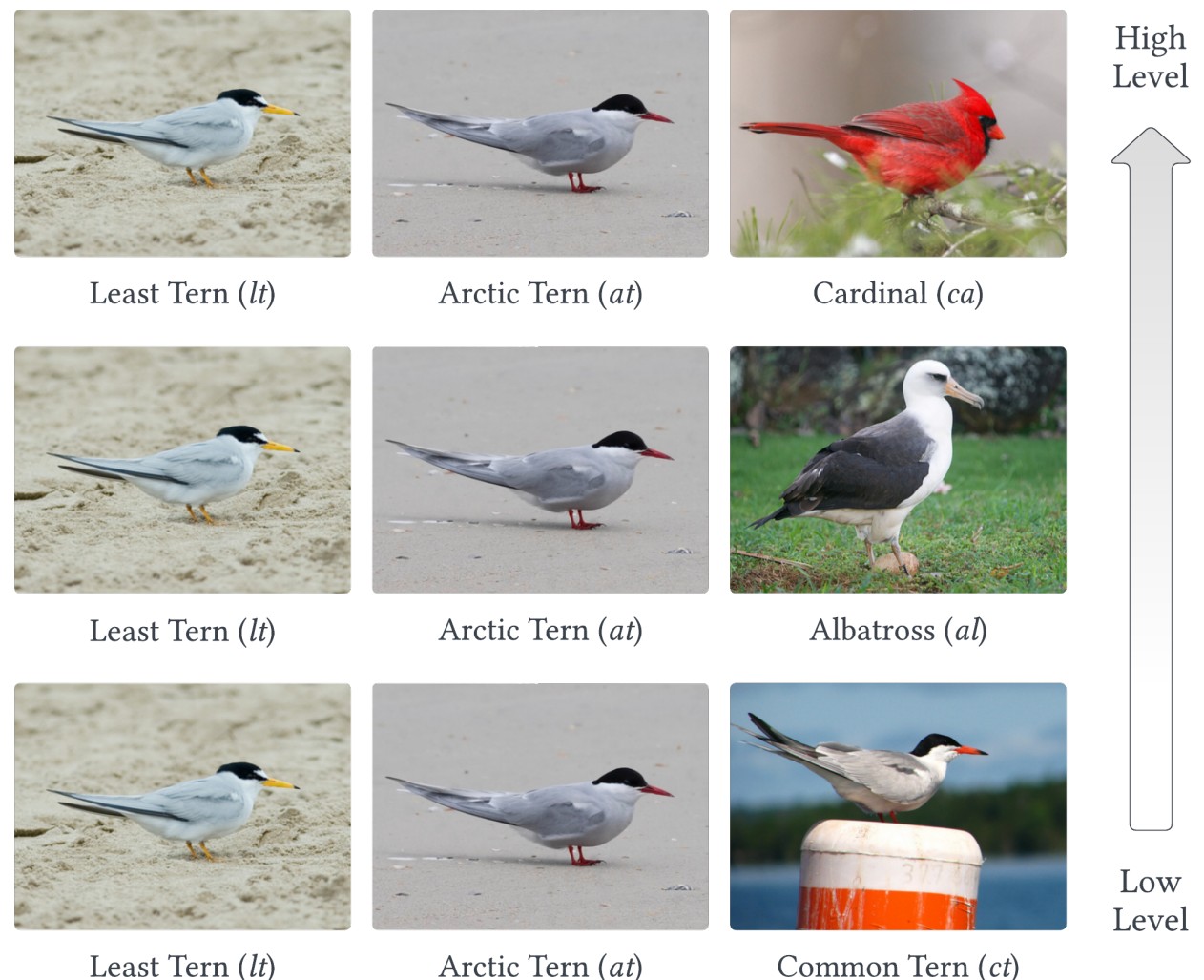

**Figure 23: Example of the three triangle queries we used to investigate the contextual bias effect. When the third bird is much different from the first two birds, crowdworkers perceive on a higher level of similarity hierarchy, thus overlooking the minor differences between the two items. When the third bird is similar to the first two birds, crowdworkers consider similarity on a lower-level hierarchy, paying more attention to the details.**

modeled noises from the crowdworker similar to bit flipping, where the answer provided by crowdworker is correct with probability $1 - q$, and incorrect with probability $q$. Pang et al. [34] considered noise as whether a query is answered or not. They assumed that a query is answered with probability $1 - q$ and not with probability $q$. Han et al. [13] pointed out that although crowdworkers give incorrect errors, some are more correct than others. For example, when the ground truth is English Foxhound, getting a label Foxhound is not totally wrong due to the hierarchical relationship between the two. Hence, they propose a new evaluation metric that measures the crowdworker's error based on how specific the label given by the crowdworker, compared to the ground truth.

## F.2 Comparison Query

Another line of work focuses on comparison query, where crowdworkers are asked to group the items by their similarity, which is based on crowdworkers' perception of them [2, 22, 25, 31, 42, 43, 45, 47]. Gomes et al. [11] showed that the wisdom

of crowds can be used for crowd clustering. Vinayak et al. [47] studied clustering algorithms that work with partially observed graphs and provided theoretical guarantees on when clustering works in such scenarios. Images in the dataset are considered as nodes in a graph. When a pair of images is deemed as similar by the crowdworkers, an edge connects the two corresponding nodes of the images. A (graph) clustering algorithm is applied to the adjacency matrix that represents the graph generated from the crowds. This work also provides experiments to demonstrate that crowdsourced clustering with a random querying strategy works more in practice. Narimanzadeh et al. [31] introduced a framework of using pairwise comparison comparison with Elo scoring to reduce the variability and bias introduced by subjectivity. They have shown that their framework outputs a better result compared to the widely used majority voting method. This work also explains why pairwise comparison is preferred over direct labeling. André et al. [2] considers the clustering task over texts instead of images.

Methods in [25 **?** ] have tried to actively select which images to be queried. However, they typically come with severe limitations, such as they need to know

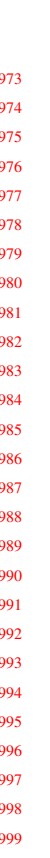

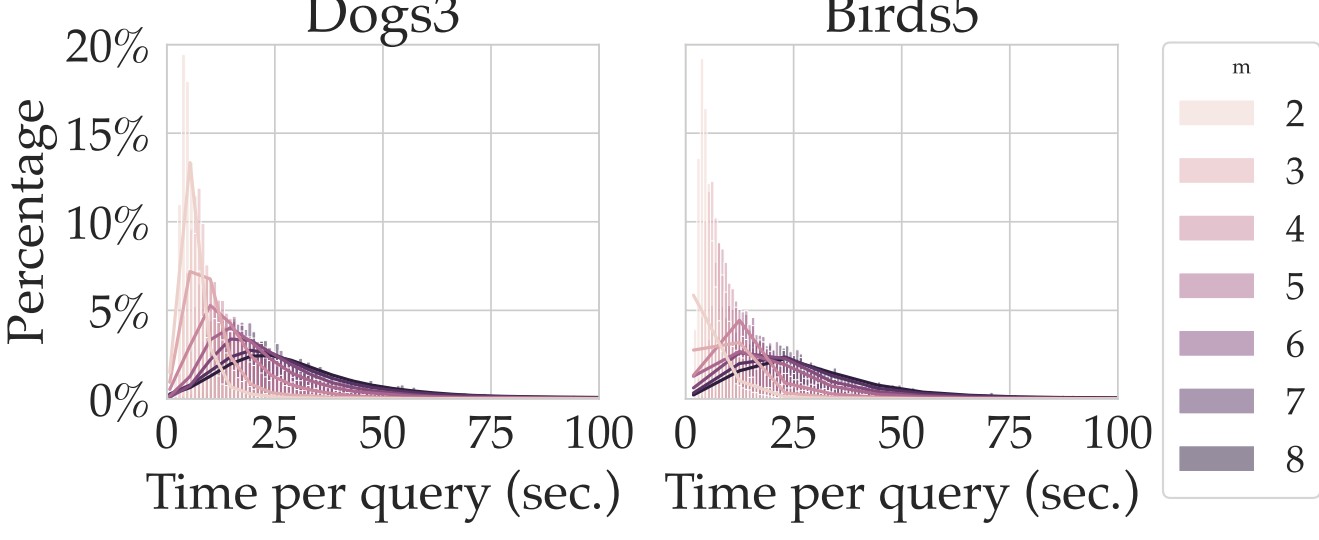

**Figure 24: Distribution of time per query for each number of items per query ($m$) on Dogs3 and Birds5 dataset.**

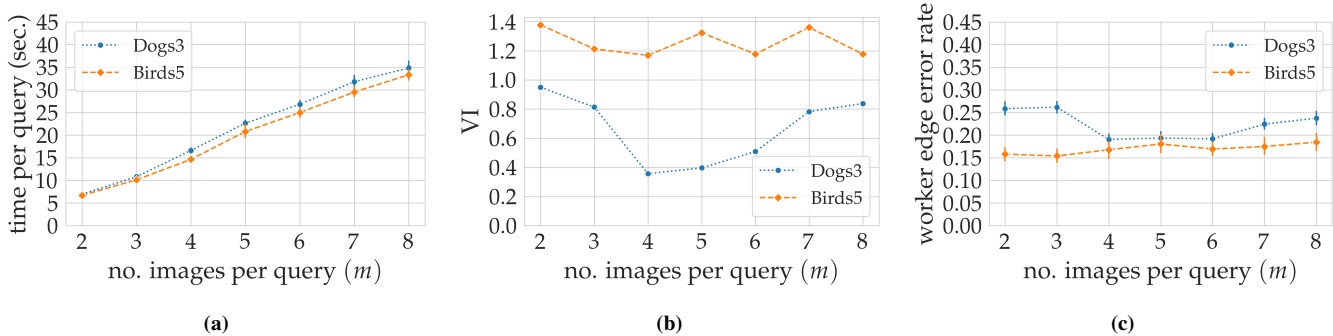

(a)                                                          (b)                                                          (c)

**Figure 25: Comparison of (a) time per query, (b) variation of information (VI), and (c) worker edge error rate between the Dogs3 and Birds5 datasets using the drag-and-drop interface, while varying the number of images per query.**

the number of clusters a priori, or they assume that crowdworkers' error is parametrized by a single scalar. Vinayak [43] present active crowdclustering, which does not rely on any unknown parameters and can recover clusters regardless of their sizes. The author provides a theoretical guarantee, under mild assumptions that crowdworkers are independent and better than random guessers, that the algorithm can recover the clusters exactly with high probability. While some simulations are provided, empirical evaluation on a real crowdsourcing platform is missing. Chen et al. [6] extends this work by implementing the algorithm and conducting experiments on AMT.

Lahouti et al. [22] proposes a method that generates clusters on the fly, instead of building an adjacency matrix and applying graph clustering on that matrix. However, they assume that crowdworkers do not make mistakes, making their method less practical. The method proposed by Vinayak and Hassibi [45], known as random triangle query, builds on top of [42] with a modification on how the question is asked. Crowdworkers in Vinayak and Hassibi [45] need to provide one of the five relationships of the three images presented: 1. All are similar, denoted by $lll$. 2. A and B are similar, $llm$. 3. A and C are similar, $lml$. 4. B and C are similar, $mll$. 5. None, $lmj$. Similar to random edge query, only a subset of all $\binom{n}{3}$ possible triplet will be queried. To model the noises, the authors present the conditional block model, which builds on top of the stochastic block model and normalizes the error probability based on the allowed configurations. The benefit of presenting three images at a time and seeking answers from 5 options is that when the budget is the same, this crowdsourcing task is more reliable than a random edge query.

### F.3 Cognitive Overload

The effect of cognitive overload, where when the number of options is increased, tasks involving comparison-based choice-making become harder and the decisions made by people become worse, has been studied extensively in the field of social psychology and information seeking [5, 7, 16, 33]. [33] and [5] discuss cognitive overload as a *"Less is More Effect"* in which people find it more difficult to draw comparisons when confronted with a large number of options. [16] study the effect in the setting of consumer behavior. The authors have found that consumers prefer to purchase from a vendor that displays fewer options. [7] identifies 4 key factors, "choice set complexity, decision task difficulty, preference uncertainty, and decision goal", that impact the effect of cognitive overload via meta-analysis in the field of consumer psychology.

### F.4 Contextual Bias

Contextual bias is the "noise" within the answers provided by crowdworkers, not due to the lack of effort or expertise, that is a function of the set of items the crowdworkers are exposed to in a query. Several workers have studied how the set of items affects the answer [2, 29]. [2] considers the clustering task over texts instead of images. It also discusses the effects of context. The result shows that having context introduced in the task is beneficial. Yet, the authors did not investigate how much context should be added. The only contextual case they have in their setting is having 10 items (text) shown at the same time.

Both Mishra and Rzeszotarski [29] and our work tries to answer the question of how the breadth of data affects the outcome of the model's result. In our work,

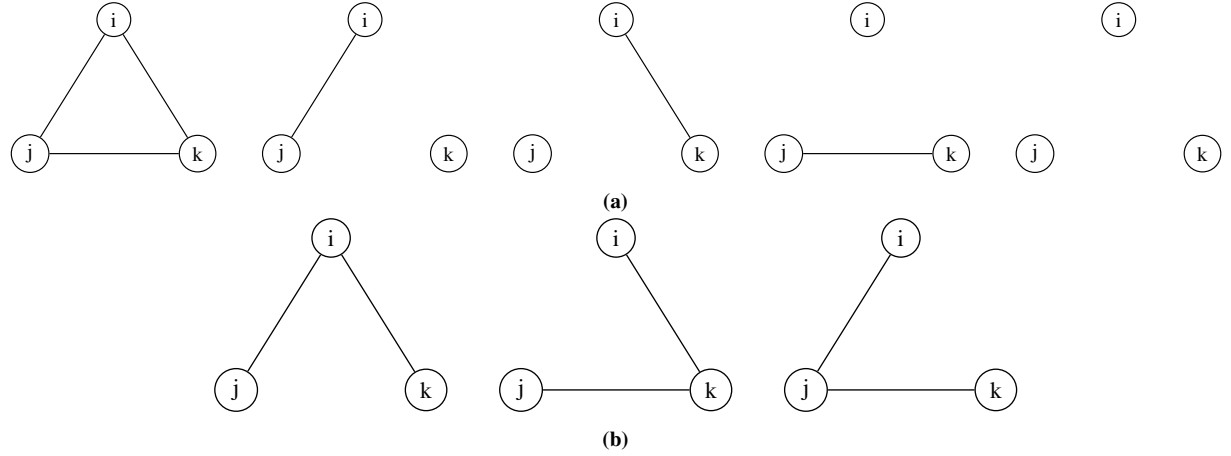

Figure 26: Configurations for a three-item query that are (a) admissible and (b) inadmissible.

| 0.56 ± 0.01 | 0.14 ± 0.01 | 0.11 ± 0.01 |
|---|---|---|
| 0.14 ± 0.01 | 0.43 ± 0.01 | 0.16 ± 0.01 |
| 0.11 ± 0.01 | 0.16 ± 0.01 | 0.62 ± 0.01 |

(a)

| 0.51 ± 0.01 | 0.09 ± 0.01 | 0.06 ± 0.01 |
|---|---|---|
| 0.09 ± 0.01 | 0.34 ± 0.01 | 0.11 ± 0.01 |
| 0.06 ± 0.01 | 0.11 ± 0.01 | 0.56 ± 0.01 |

(b)

| 0.47 ± 0.01 | 0.07 ± 0.01 | 0.04 ± 0.01 |
|---|---|---|
| 0.07 ± 0.01 | 0.31 ± 0.01 | 0.08 ± 0.01 |
| 0.04 ± 0.01 | 0.08 ± 0.01 | 0.53 ± 0.01 |

(c)

| 0.44 ± 0.01 | 0.05 ± 0.01 | 0.03 ± 0.01 |
|---|---|---|
| 0.05 ± 0.01 | 0.28 ± 0.01 | 0.06 ± 0.01 |
| 0.03 ± 0.01 | 0.06 ± 0.01 | 0.52 ± 0.01 |

(d)

| 0.42 ± 0.01 | 0.04 ± 0.01 | 0.03 ± 0.01 |
|---|---|---|
| 0.04 ± 0.01 | 0.26 ± 0.01 | 0.05 ± 0.01 |
| 0.03 ± 0.01 | 0.05 ± 0.01 | 0.51 ± 0.01 |

(e)

| 0.4 ± 0.0 | 0.04 ± 0.01 | 0.02 ± 0.01 |
|---|---|---|
| 0.04 ± 0.01 | 0.24 ± 0.01 | 0.04 ± 0.01 |
| 0.02 ± 0.01 | 0.04 ± 0.01 | 0.5 ± 0.01 |

(f)

| 0.39 ± 0.0 | 0.03 ± 0.0 | 0.02 ± 0.0 |
|---|---|---|
| 0.03 ± 0.0 | 0.22 ± 0.01 | 0.04 ± 0.01 |
| 0.02 ± 0.0 | 0.04 ± 0.01 | 0.5 ± 0.01 |

(g)

Figure 27: Empirical edge density matrices obtained from simulation using the empirical edge density obtained from drag-and-drop interface when $m = 2$. Matrices (a)-(g) correspond to the Dogs3 dataset, with $m$ ranging from 2 to 8. The ± indicates the confidence intervals calculated as described in Section 5.2.

however, the breadth concerns the set of items being shown to crowdworkers, rather than being used by the model. For the granularity aspect, Mishra and Rzeszotarski [29] considers granularity as the level of detail used (by the model) to explain a model's decision. Conversely, we treat granularity as the level of detail used by crowdworkers to make their decision. We could consider our work as a reverse version of [29], in a way such that crowdworkers in our work are the

explainable model in their work (although the crowdworkers in our work do not explain how they make the decision).

Received 20 February 2007; revised 12 March 2009; accepted 5 June 2009

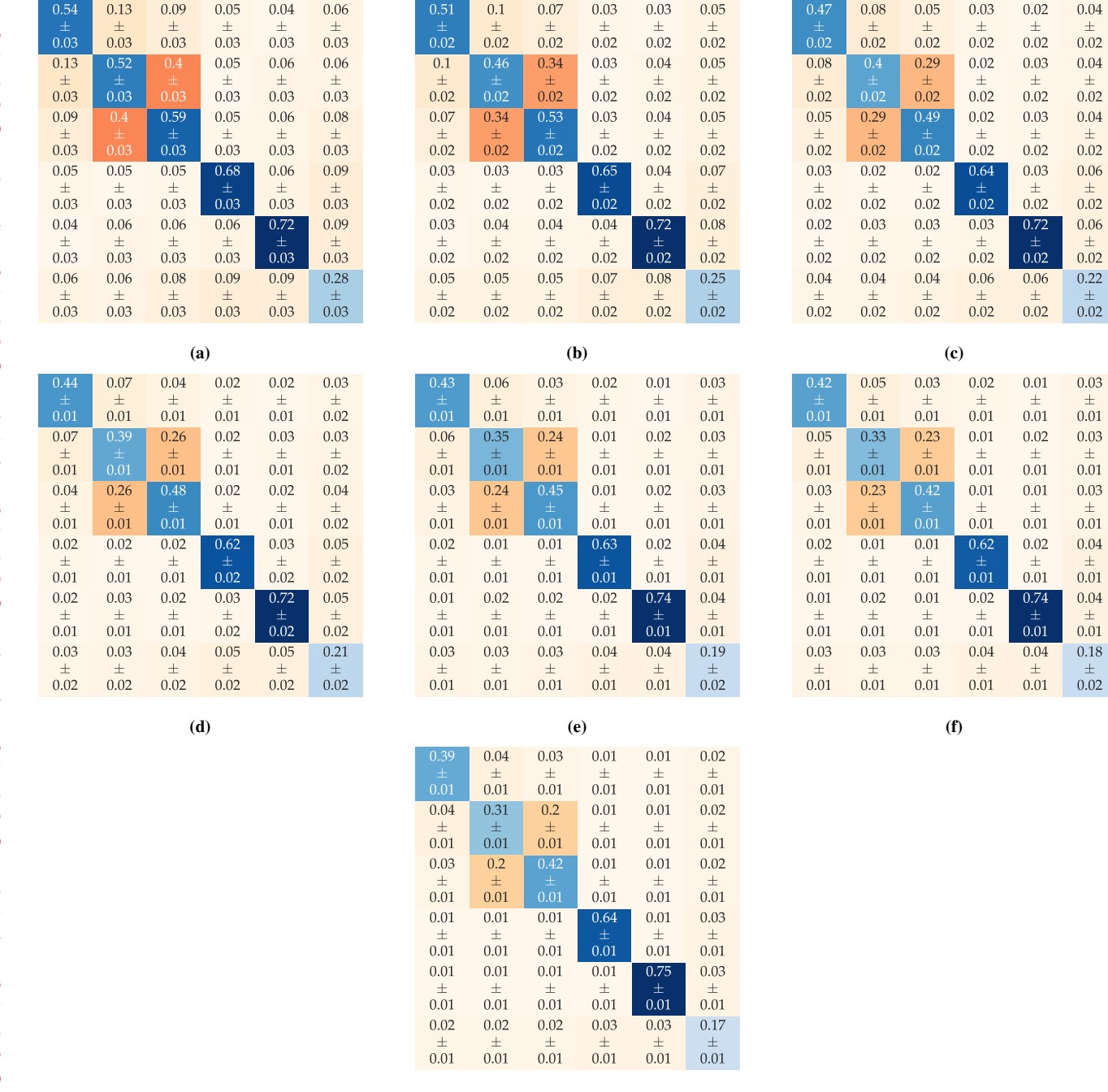

Figure 28: Empirical edge density matrices obtained from simulation using the empirical edge density obtained from drag-and-drop interface when $m = 2$. Matrices (a)-(g) correspond to the Birds5 dataset, with $m$ ranging from 2 to 8. The ± indicates the confidence intervals calculated as described in Section 5.2.

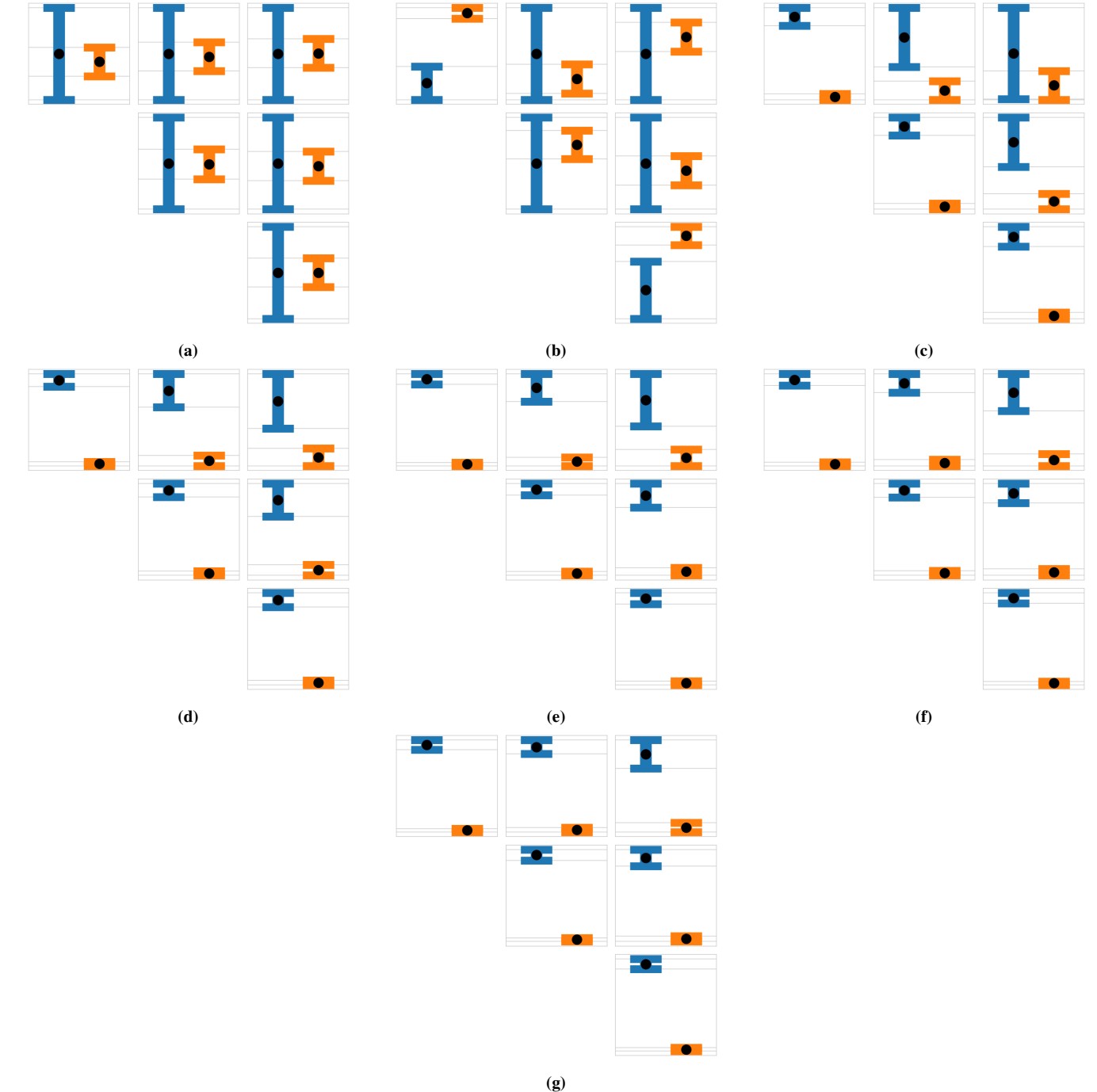

**Figure 29: Visualization of the concentration bounds for the entries in the edge density matrix obtained from crowdsourcing experiments (blue) and simulations (orange). Matrices (a)-(g) compare the bounds on each entry with $m$ ranging from 2 to 8 and Dogs3 dataset. It can be seen that as $m$ increases, more and more entries contain bounds that do not overlap. This indicates that the edge density matrix predicted by CBM does not match with our empirical observation.**

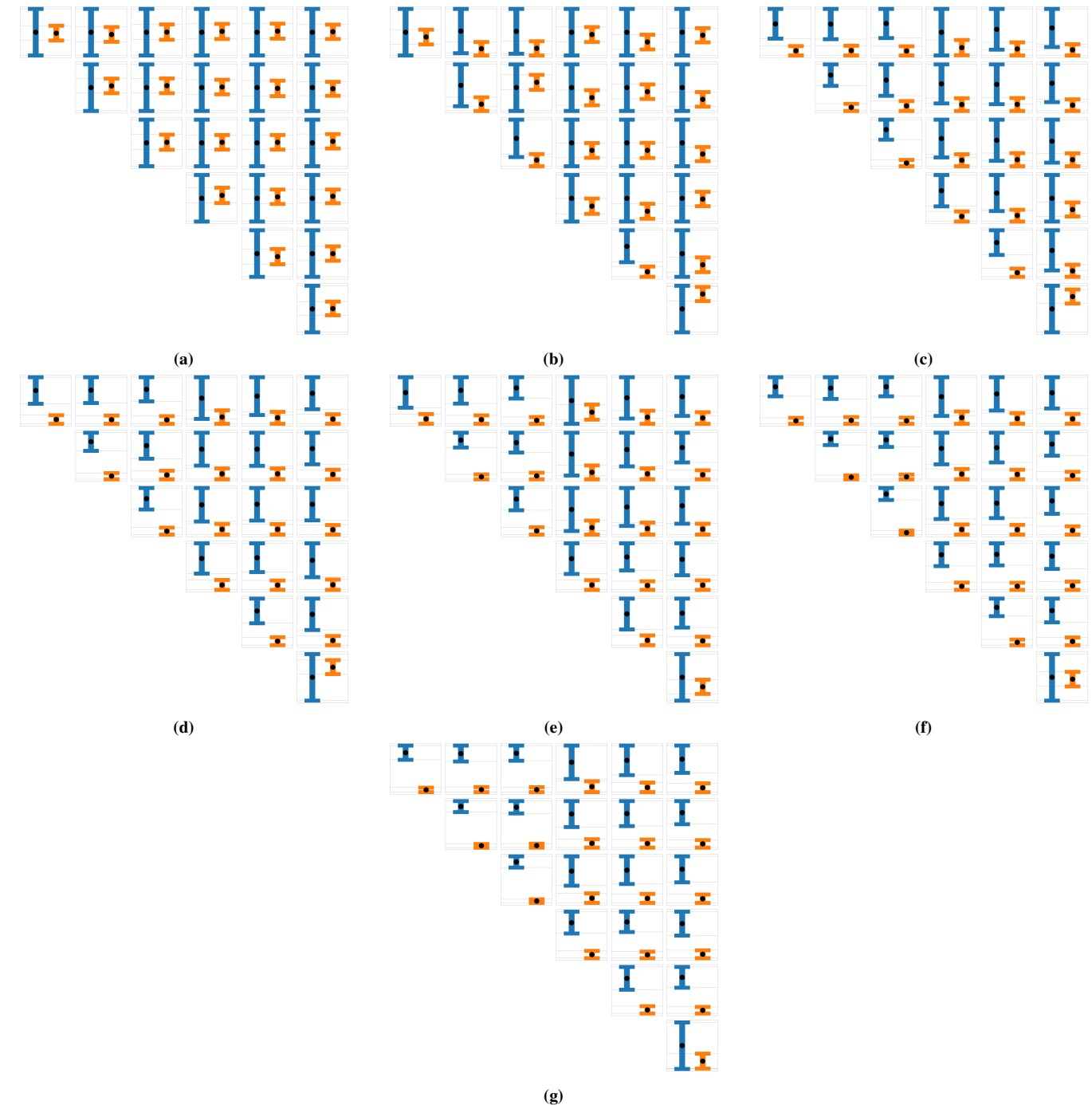

**Figure 30: Visualization of the concentration bounds for the entries in the edge density matrix obtained from crowdsourcing experiments (blue) and simulations (orange). Matrices (a)-(g) compare the bounds on each entry with $m$ ranging from 2 to 8 and Birds5 dataset. It can be seen that as $m$ increases, more and more entries contain bounds that do not overlap. This indicates that the edge density matrix predicted by CBM does not match with our empirical observation.**

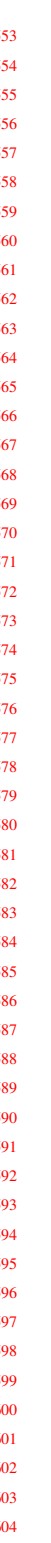
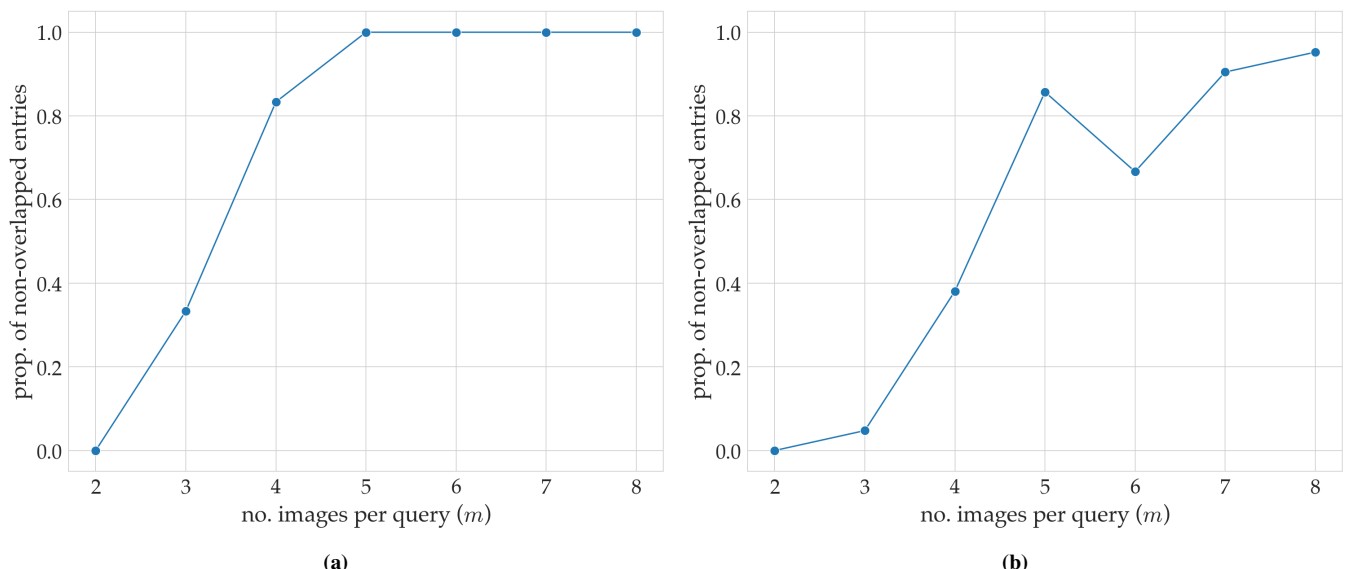

(a)

(b)

**Figure 31: Proportion of non-overlapped entries to the total number of unique entries of the edge density matrix as a function of $m$ for (a) Dogs3 and (b) Birds5 dataset. As $m$ increases, the number of entries contain bounds that do not overlap also increase.**

