# OpenReview forum: "Query Design for Crowdsourced Clustering: Effect of Cognitive Overload and Contextual Bias"
_ACM.org/TheWebConf/2025/Conference — WWW 2025 Oral_

### Official Review · Reviewer_7BSi · 2024-11-24

**Novelty:** 4
**Technical Quality:** 4

**Review:**

The paper introduces contextual bias as a critical factor in crowdsourced clustering tasks, which challenges existing noise models and offers an innovative perspective.
The authors provide a rigorous investigation, including empirical studies on cognitive overload and contextual bias, supported by both experiments and simulations.

Theoretical Underpinning: While the empirical analysis is strong, the theoretical modeling of contextual bias could be expanded to provide deeper insights into the observed patterns.
Scalability: The proposed methods for multi-item queries might face practical scalability issues in larger datasets, which is not thoroughly discussed.
Limited Generalizability: The study primarily focuses on image datasets, and its applicability to other domains (e.g., text clustering) is not explored.
Interface Comparisons: The discussion comparing drag-and-drop versus radio-button interfaces lacks depth in terms of their implications on worker engagement and performance.

**Questions:**

Can you elaborate on how contextual bias might differ in other types of crowdsourced tasks, such as text-based clustering or semantic annotation?
What are the computational trade-offs when scaling the multi-item query design to datasets with significantly larger sizes and more clusters?
Could the findings related to diminishing returns at 4-5 items per query be generalized across different cultures or demographic groups of crowdworkers?
How might incorporating machine learning models (e.g., pretrained vision transformers) help mitigate contextual bias or cognitive overload in future implementations?
Were there any notable trends in worker feedback regarding task difficulty or preference for different interfaces?

**Reviewer Confidence:**

2: The reviewer is willing to defend the evaluation, but it is likely that the reviewer did not understand parts of the paper

**Scope:**

2: The connection to the Web is incidental, e.g., use of Web data or API

---

### Official Review · Reviewer_HSUA · 2024-11-29

**Novelty:** 5
**Technical Quality:** 5

**Review:**

**Summary**

The paper studies the cognitive effects in crowdsourced clustering tasks. The paper conducts experiments to validate the existence of cognitive overload and contextual bias. The simulations show that the existing model does not capture the cognitive overload effect.

**Strength**

The paper considers an important question in crowdsourcing, the query design problem. The experiments are rigorously designed and analysis is theoretically justified. The findings suggest that behavioral patterns may lead to noise, which current models may not capture.

**Weakness**

The results are not surprising. I found two issues with the simulation not convincing enough.

* The simulation fixes a set of error probabilities. E.g. for the simulation with empirical frequency, the matrix is estimated from the case with m=2, which makes the simulation result less surprising. However, I assume the purpose of such models is for ground truth recovery. For ground truth discovery, the error rates are assumed to be the actual empirical error rate with the associated experimental setup, not for m=2.

* Following the previous point, the paper suggests new models are needed to capture the behavioral noise, which is not fully justified. How does behavioral noise affect the performance of the models? e.g. does it make ground truth recovery to be somewhat harder or biased?


**Minor comments**

* Line 622 exceeding linewidth.

**Questions:**

I might be missing something, but the paper says

*"We use simulations to demonstrate that existing models cannot fully capture crowdworker errors, especially those due to contextual bias. The conditional block model (CBM) proposed in [45] has the potential to incorporate contextual bias in a crowdsourced clustering setting. "*

which I'm not fully following. How does CBM has the potential for contextual bias?

**Reviewer Confidence:**

3: The reviewer is confident but not certain that the evaluation is correct

**Scope:**

4: The work is relevant to the Web and to the track, and is of broad interest to the community

---

### Official Review · Reviewer_qctG · 2024-11-29

**Novelty:** 6
**Technical Quality:** 6

**Review:**

#### Quality
The paper presents a well-structured and comprehensive study on the impact of query design in crowdsourced clustering, specifically focusing on cognitive overload and contextual bias. The experiments are meticulously designed, using standard datasets and appropriate evaluation metrics, which lends credibility to the findings. The inclusion of both empirical studies and simulation results provides a robust foundation for the conclusions drawn.

#### Significance
The findings have practical implications for designing crowdsourcing tasks, particularly in scenarios requiring clustering of large datasets. The identification of an optimal query size (around 4-5 items) and the acknowledgment of contextual bias can lead to improved task designs and more accurate clustering results.

### Pros
1. **Comprehensive Study**: The paper thoroughly investigates cognitive overload and contextual bias, providing both empirical and simulation-based evidence.
2. **Methodological Rigor**: The use of standard datasets and appropriate evaluation metrics enhances the validity of the results.
3. **Novel Insights**: The exploration of contextual bias and its impact on clustering outcomes is a significant contribution to the field.
4. **Practical Implications**: The findings suggest practical guidelines for designing crowdsourcing tasks, which can improve the efficiency and accuracy of clustering.

### Improvement
1. **Query Repetition**: The lack of query repetition may introduce variability in the results. Including redundancy could provide a more reliable assessment of worker accuracy.
2. **Interface Design**: While the drag-and-drop interface is scalable, its usability could be improved with more extensive user testing, especially for new users.

**Questions:**

1. How was contextual bias quantified, and how does its impact compare to cognitive overload in influencing clustering outcomes?

2. Have the findings been tested or considered for generalization across different datasets or domains, and are there plans for such exploration?

3. What modifications do the authors propose for existing noise models like CBM to better capture observed noise patterns, and are there alternative models being considered?

**Reviewer Confidence:**

2: The reviewer is willing to defend the evaluation, but it is likely that the reviewer did not understand parts of the paper

**Scope:**

1: The work is irrelevant to the Web

---

### Official Review · Reviewer_tzg5 · 2024-12-02

**Novelty:** 5
**Technical Quality:** 5

**Review:**

## Summary:

The paper explores the trade-off between the number of items presented per query to crowdworkers and the accuracy of their responses, highlighting how cognitive overload leads to diminishing returns. Additionally, the paper identifies the presence of contextual bias, where the likelihood of grouping items is influenced not only by their similarity but also by the other items present in the query. Additionally, they demonstrate through simulations that the conditional block model (CBM) fails to capture errors arising from contextual bias.

## Strengths:

S1. The paper is well-structured and presents its findings in a clear and concise manner.

S2. The findings provide valuable insights for designing effective crowdsourced clustering systems, particularly highlighting the importance of considering cognitive overload and contextual bias when determining number of items in a query.

S3. The experiment in the sources offer compelling evidence to support the hypotheses regarding cognitive overload and contextual bias.

S4. Comparison between the Conditional Block Model (CBM) and human behavior reveals the model's inability to capture contextual bias.

## Weaknesses:

W1. The study primarily focuses on two specific datasets: Dogs3, comprising images of three dog breeds, and Birds5, containing images of five bird species. Future research should investigate the generalizability of these findings to other types of datasets with varying levels of complexity and ambiguity.

W2. The demonstration of contextual bias relies on a single set of comparisons involving Least Terns, Arctic Terns, and a third varying species. Further investigation is needed to ascertain whether similar patterns of bias emerge with different species or object categories.

W3. The study does not address the potential impact of the number of classes in the datasets on the observed results, particularly in relation to cognitive overload.

**Questions:**

Q1. Do similar bias patterns appear with other species or object categories, besides the ones shown in the experiments?

Q2. Have you considered how the number of classes might affect cognitive overload?

**Reviewer Confidence:**

3: The reviewer is confident but not certain that the evaluation is correct

**Scope:**

3: The work is somewhat relevant to the Web and to the track, and is of narrow interest to a sub-community